# ParB dynamics and the critical role of the CTD in DNA condensation unveiled by combined force-fluorescence measurements

Julene Madariaga-Marcos[1], Cesar L Pastrana[1], Gemma LM Fisher[2], Mark Simon Dillingham[2], Fernando Moreno-Herrero[1]*

[1]Department of Macromolecular Structures, Centro Nacional de Biotecnología, Consejo Superior de Investigaciones Científicas, Madrid, Spain; [2]DNA:Protein Interactions Unit, School of Biochemistry, Faculty of Life Sciences, University of Bristol, Bristol, United Kingdom

**Abstract** *Bacillus subtilis* ParB forms multimeric networks involving non-specific DNA binding leading to DNA condensation. Previously, we found that an excess of the free C-terminal domain (CTD) of ParB impeded DNA condensation or promoted decondensation of pre-assembled networks (Fisher et al., 2017). However, interpretation of the molecular basis for this phenomenon was complicated by our inability to uncouple protein binding from DNA condensation. Here, we have combined lateral magnetic tweezers with TIRF microscopy to simultaneously control the restrictive force against condensation and to visualise ParB protein binding by fluorescence. At non-permissive forces for condensation, ParB binds non-specifically and highly dynamically to DNA. Our new approach concluded that the free CTD blocks the formation of ParB networks by heterodimerisation with full length DNA-bound ParB. This strongly supports a model in which the CTD acts as a key bridging interface between distal DNA binding loci within ParB networks.
DOI: https://doi.org/10.7554/eLife.43812.001

**\*For correspondence:**
fernando.moreno@cnb.csic.es

**Competing interests:** The authors declare that no competing interests exist.

## Introduction

Faithful segregation of bacterial chromosomes is mediated by partition systems that are classified depending on the type of NTPase involved (*Gerdes et al., 2010*). Type I partition systems, also known as ParABS include an ATPase motor protein, ParA, responsible for the movement of the replicated chromosomes to the distal pole of the cell, a DNA binding protein ParB, and a centromere-like DNA sequence, *parS* (*Funnell, 2016*). This type of partition system has been described in many bacteria, including *Bacillus subtilis*, where it is involved in chromosome segregation and sporulation (*Mierzejewska and Jagura-Burdzy, 2012*) (*Yamaichi and Niki, 2000*) (*Sharpe and Errington, 1996*) (*Lin and Grossman, 1998*) (*Lee and Grossman, 2006*). Importantly, the discovery that *B. subtilis* Structural Maintenance of Chromosome (SMC) proteins are loaded by ParB at *parS* sites in vivo and that *parS* sites act as condensation centers highlights the critical role played by ParB in chromosome organisation (*Gruber and Errington, 2009*) (*Sullivan et al., 2009*) (*Umbarger et al., 2011*) (*Broedersz et al., 2014*) (*Taylor et al., 2015*). Interestingly, apart from the specific binding to partition sites, ParB proteins can also bind non-specifically to DNAs spreading several kilobases around *parS* sites (*Breier and Grossman, 2007*) (*Murray et al., 2006*). This spreading is achieved by a limited number of ParB protomers, which has been rationalised by models of ParB-mediated bridging and condensation of DNA supported by in vitro analysis (*Graham et al., 2014*) (*Taylor et al., 2015*) (*Fisher et al., 2017*) (*Broedersz et al., 2014*) (*Sanchez et al., 2015*). The generation of these 'ParB

networks' in vivo necessarily requires the presence of specific and non-specific DNA-binding domains in ParB as well as multimerisation interfaces between ParB proteins. Recent work has begun to unpick the structural basis for formation of these networks and has implicated both the N- and C-terminal domains in forming and maintaining bridging interactions (*Leonard et al., 2004*) (*Fisher et al., 2017*) (*Song et al., 2017*) (*Chen et al., 2015*).

In our previous work (*Fisher et al., 2017*), we studied the role of the C-terminal domain (CTD) of ParB in its multimerisation and DNA condensation. In particular, we confirmed its dimeric stoichiometry and identified a novel intermolecular non-specific DNA-binding site across the positively-charged β-sheet face of the CTD of ParB, which was critical for DNA condensation. We also showed that the presence of an excess of the CTD prevented DNA condensation by ParB and induced disruption of ParB networks in vitro and in vivo. This led us to propose three possible scenarios to explain CTD-mediated condensation inhibition (*Figure 1*). One possibility is that the CTD simply competes for the binding sites in the DNA molecule (*Figure 1B*, DNA-binding competition model).

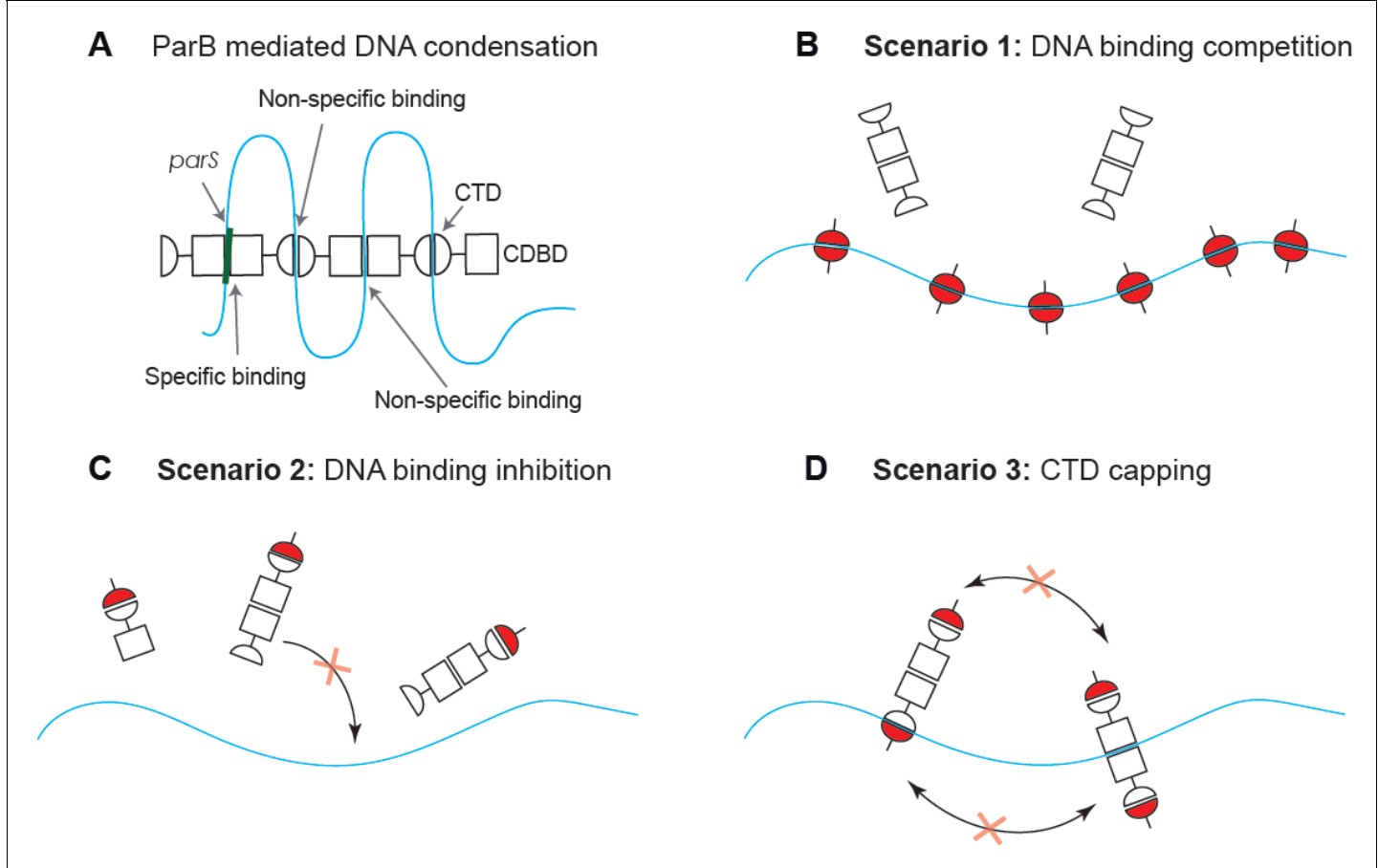

**Figure 1.** Possible scenarios for CTD-induced decondensation of ParB-DNA networks. (**A**) Model for ParB network formation and condensation via ParB-DNA and ParB-ParB interactions. ParB monomers comprise a central DNA-binding domain (CDBD) with specific and possibly non-specific DNA-binding activities and a carboxy-terminal domain (CTD) with non-specific DNA-binding capacity. The amino terminal domain (NTD) is not represented on the scheme for clarity. (**B**) In scenario 1, the CTD can bind dsDNA competing for the DNA-binding sites, displacing full length ParB and therefore de-condensing the DNA. (**C**) In scenario 2, the CTD can exchange with full length ParB forming heterodimers in free solution that are inactive so can no longer exchange with the ParB-DNA network. (**D**) In scenario 3, the CTD forms heterooligomers with DNA-bound ParB which retain DNA-binding activity but are not able to condense DNA because bridging interactions are 'capped' by the CTD. Possible scenarios for CTD-induced decondensation of ParB-DNA networks.

DOI: https://doi.org/10.7554/eLife.43812.002

In this way, the CTD would displace ParB from the DNA and thus prevent DNA condensation, provided the CTD cannot condense DNA on its own. Another possible scenario is that the CTD is able to heterodimerise with free ParB in solution and these heterodimers are deficient for DNA binding, which directly impedes condensation (*Figure 1C*, DNA-binding inhibition model). Finally, it is possible that the CTD interacts with DNA-bound ParB, forming heterodimers that cannot condense but which may remain bound to DNA (*Figure 1D*, CTD capping model). This deficiency in condensation could be caused by the CTD 'capping' the ParB dimerisation interfaces.

Our previous experiments with CTD variants defective for DNA binding suggested that ParB-network disruption was unlikely to be mediated by direct competition for DNA binding sites (*Fisher et al., 2017*). However, because the ParB binding and DNA condensation signals were tightly coupled (*Taylor et al., 2015*) (*Fisher et al., 2017*), it was not possible to separate ParB binding from condensation or to correlate both measurements in parallel. This precluded a direct test of our favoured capping model, which predicts (uniquely) that the free CTD will inhibit DNA condensation but not DNA binding. For instance, in ensemble experiments using electrophoretic mobility shift assays and ParB-protein-induced fluorescence enhancement assays, the signal reported is concomitant to protein binding and condensation (*Taylor et al., 2015*). The same happens in single-molecule stretch-flow fluorescence experiments (*Graham et al., 2014*) requiring the use of condensation-deficient mutants to determine the kinetic parameters of DNA binding (*Song and Loparo, 2015*). Additionally, in stretch-flow experiments, condensation is restricted to the free end of the molecule due to a decreasing force gradient along the DNA. Magnetic Tweezers (MT) experiments overcome some of these difficulties in that the applied force is uniform and can be used to prevent DNA condensation. It is therefore possible to measure DNA condensation when the force is decreased to permissive levels (*Taylor et al., 2015*). Nonetheless, in MT it is not possible to visualise and determine the degree of protein binding to DNA molecules. Here, we have combined lateral MT with total internal reflection fluorescence (TIRF) microscopy (*Madariaga-Marcos et al., 2018*) to simultaneously control the force applied to DNA and to visualise ParB proteins by fluorescence. This allowed us to directly challenge the three possible scenarios to explain the condensation inhibition mechanism of the CTD of ParB. In addition, the combination of our MT-TIRF device with multilaminar flow technology facilitated the rapid exposure of DNA to different protein concentrations and provided a method to precisely capture the kinetics of binding and unbinding events. The experimental platform used here allowed us to confirm that the inhibition of ParB condensation by the free CTD occurs without displacement of full length ParB from DNA. Our data indicates that the free CTD of ParB blocks the formation of ParB networks via a dominant negative effect on a key protein-protein bridging interface.

## Results and discussion

### A new method for monitoring ParB DNA binding under conditions that are non-permissive for DNA condensation

We laterally stretched single ~24 kbps DNA molecules lacking the *parS* sequence (see Materials and methods) (*Figure 2A*) using a home-built magnet holder that can pull DNA molecules from one side (*Madariaga-Marcos et al., 2018*). This device can apply forces of ~1 pN on 1 μm beads, which is sufficient to prevent condensation of DNA upon introduction of ParB. We produced a fluorescent variant of ParB labelled with Alexa Fluor 488 (ParB$_2$$^{S68C-AlexaFluor488}$, hereafter ParB$_2$$^{AF}$), which retained wild-type DNA-binding properties in our established bulk assays (*Figure 2—figure supplement 1*). This allowed us to visualise protein binding while stretching the DNA at forces non-permissive for condensation (*Figure 2B*). Injection of the protein in the flow cell resulted in a gradual increase of fluorescence concomitant to the arrival and non-specific binding of the protein to the DNA (*Video 1*). Note, however, that sometimes a fraction of the DNA closer to the magnetic bead remained invisible in the fluorescence signal (*Video 2*). This is as expected and is due to the limited penetration of the excitation light produced by TIRF, the tilting angle of the DNA ($\leq 5^0$), the attachment point of the DNA at the bead, and the fixed bead orientation under the magnetic field.

The integrated intensity in the area of a DNA molecule remained constant for the longest time points tested (2 min) in the absence of an oxygen scavenger system, suggesting a continuous and fast exchange of ParB proteins with the surrounding media (*Figure 2C*). The integrated intensity

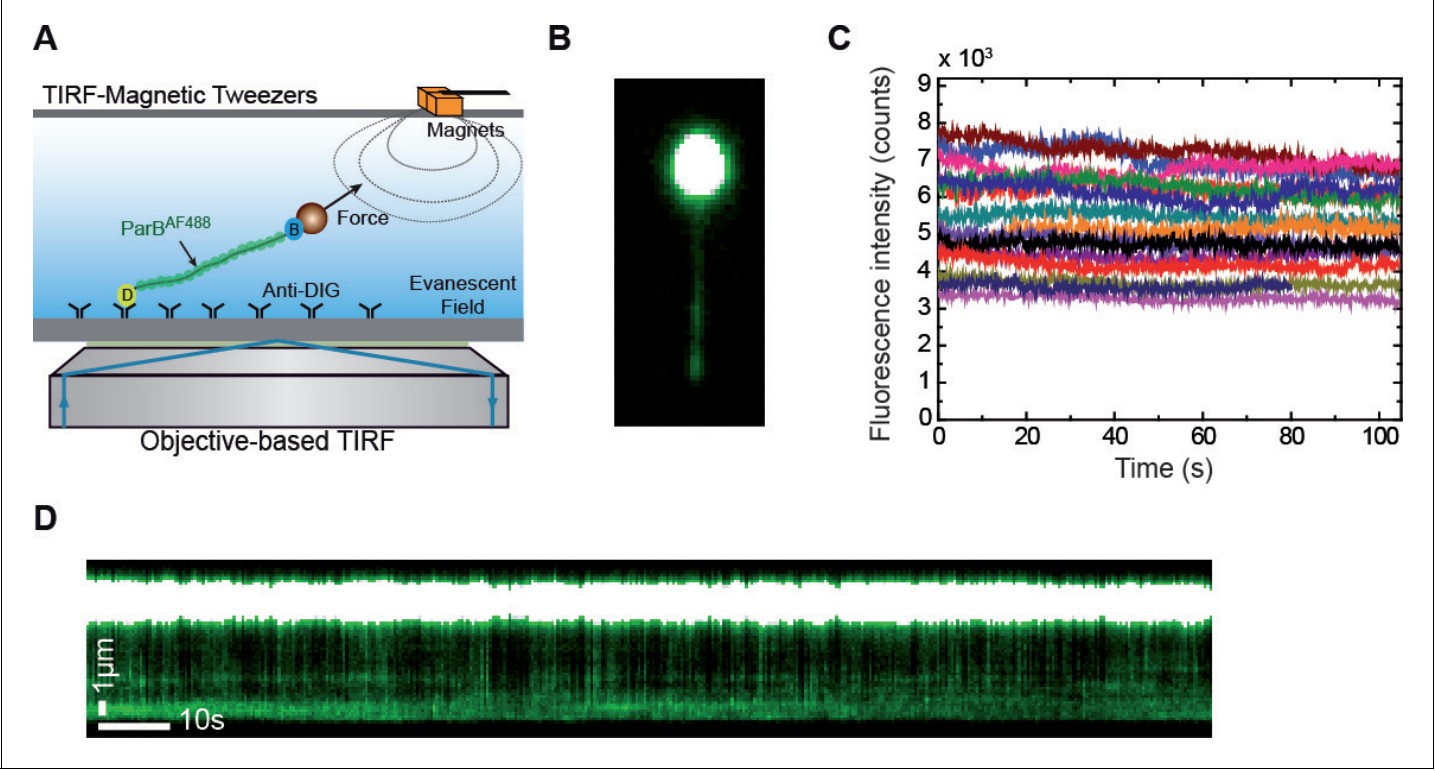

**Figure 2.** Combined lateral MT and TIRF microscopy to study ParB-DNA interactions. (A) Cartoon of the MT-TIRF setup used to visualise ParB-DNA interactions at the single-molecule level with controlled external force. A magnet pulls laterally on the distal end of a DNA molecule which is anchored to the coverslip. Fluorescently labelled ParB[AF] is excited in TIRF mode using 488 nm laser light and the emitted light is collected by an EM-CCD (*Madariaga-Marcos et al., 2018*). (B) TIRF image showing a laterally stretched DNA molecule under a force of 1 pN and in the presence of 500 nM ParB[AF]. Beads showed fluorescence due to additional binding of DNA molecules, which are further labelled by non-specifically bound ParB[AF] proteins. (C) Intensity of several DNA molecules (n = 15) in the presence of 500 nM ParB[AF] as a function of time. Even though the integrated intensity changes from molecule to molecule, intensity remains constant for more than 100 s, suggesting a dynamic and fast exchange between DNA-bound ParB[AF] and free ParB[AF] in the media. (D) Kymograph from the same experimental data shown in (B) highlighting changes in intensity along the DNA molecule through the entire experiment supporting the continuous exchange of the protein. Kymographs varied from molecule to molecule. All experiments were conducted in ParB reaction buffer supplemented with 4 mM $Mg^{2+}$.

DOI: https://doi.org/10.7554/eLife.43812.003

The following source data and figure supplements are available for figure 2:

**Source data 1.** Fluorescence intensity of 15 DNA molecules in the presence of 500 nM ParB[AF] as a function of time.
DOI: https://doi.org/10.7554/eLife.43812.006

**Figure supplement 1.** Purification and activity assays of ParB[AF] and CTD[AF].
DOI: https://doi.org/10.7554/eLife.43812.004

**Figure supplement 2.** Control experiments showing increasing intensity due to ParB[AF] binding and constant intensity due to protein exchange.
DOI: https://doi.org/10.7554/eLife.43812.005

differs between molecules due to the fraction of DNA imaged, which depends on the anchoring point of the DNA on the bead. This was further confirmed in kymographs taken along the DNA that showed variations in fluorescence intensity with time (*Figure 2D*, *Video 1*). As expected, we did observe a gradual decrease of fluorescence intensity when the protein was removed from free solution by flowing buffer alone (*Video 3* and *Figure 2—figure supplement 2A*). Control experiments crosslinking ParB with formaldehyde indicated a photobleaching half-life time for Alexa Fluor of around 28 s, much shorter than the duration of our measurements (*Figure 2—figure supplement 2B*). The rapid exchange of the protein was further confirmed in fluorescence recovery after photobleaching (FRAP) experiments. They consisted of photo-bleaching all ParB[AF] molecules included in the visualisation area inside the TIRF field by using a high-laser power pulse, and then monitoring the fluorescence recovery of the protein-DNA complex in the presence of 250 nM ParB[AF] (*Video 4*

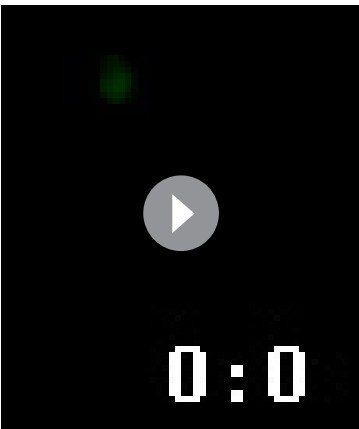

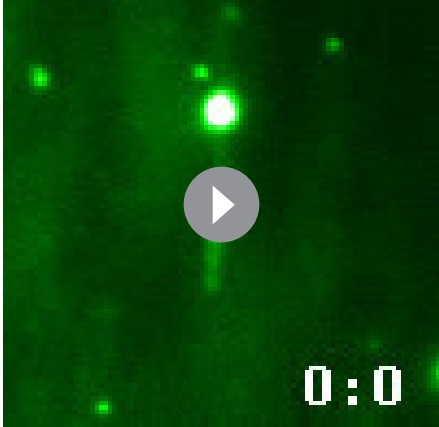

**Video 1.** Movie showing that ParB[AF] binding increases the intensity along a DNA molecule. Flow rate of 250 µl/min. Movie is 5X accelerated.
DOI: https://doi.org/10.7554/eLife.43812.007

**Video 3.** Movie showing that the intensity of DNA coated with 250 nM ParB[AF] decays flowing buffer (250 µl/min). Movie is 5X accelerated.
DOI: https://doi.org/10.7554/eLife.43812.009

and *Figure 2—figure supplement 2C*). Altogether, these experiments demonstrate the dynamic interaction of ParB with DNA under forces that are non-permissive for condensation.

## Association and dissociation kinetics of ParB measured at non-permissive forces for condensation

Prompted by the observation of fast ParB[AF] exchange, we attempted to elucidate the kinetics of the non-specific binding and unbinding of ParB[AF]. The simplest approach would be to repeatedly exchange protein-containing and protein-free buffer and to monitor changes in the fluorescence signal over time. However, the arrival of the protein at the site of interest in single-channel flow cells is gradual, typically requires tens of seconds to reach equilibrium, and it is difficult to assign arrival times (*Gollnick et al., 2015*; *Liu et al., 2013*). Although this effect can be minimised by using high flow velocities, it nevertheless would affect the measured kinetics of protein association. Moreover, flowing at high rates increases the force applied on the DNA as a result of the drag force on the bead and can lead to structural distortions of the DNA duplex, ultimately interfering with protein binding (*Xiao et al., 2011*).

In order to reduce reagent exchange times and the contribution of the flow on the total force on the DNA, we developed a fast buffer exchange system based on multilaminar flow cells (*Liu et al., 2013*) (*Tan et al., 2007*) (*Brouwer et al., 2018*). We used flow cells with two inlets, where two reagents are introduced (i.e. protein and buffer), and a single outlet (*Figure 3A*). The dimensions of our flow cells and the flow rates we used ($\leq$500 µl/min) ensure a laminar flow regime under which the two reagents do not significantly mix (Materials and methods) (*Brewer and Bianco, 2008*). Fast alternation between buffer and protein was achieved by varying the flow rates of the syringes (190 µl/min and 10 µl/min) and maintaining a constant

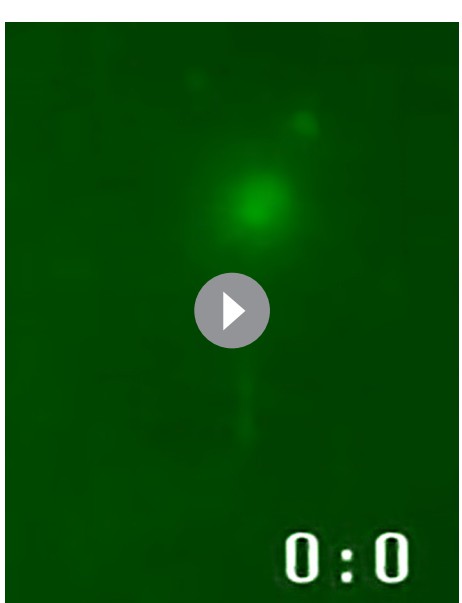

**Video 2.** Movie showing that intensity remains constant when imaging a ParB[AF] coated DNA molecule (250 nM). Movie is 5X accelerated.
DOI: https://doi.org/10.7554/eLife.43812.008

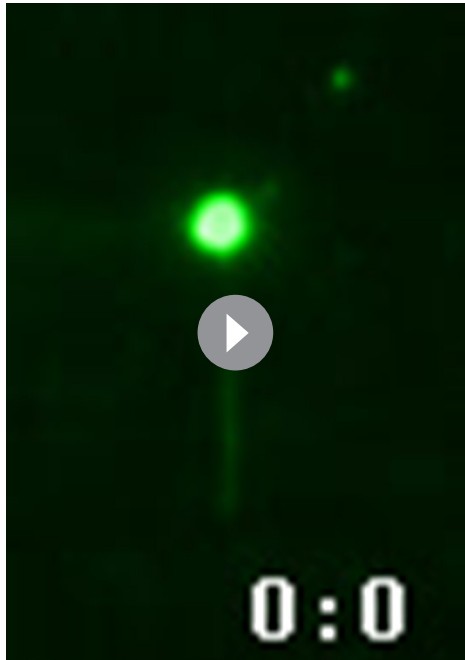

**Video 4.** Movie showing that the intensity of DNA coated with 250 nM ParB$^{AF}$ is recovered in FRAP experiment. Movie is 5X accelerated.
DOI: https://doi.org/10.7554/eLife.43812.010

flow rate in the central channel (200 µl/min). This reduced reagent switching times, and allowed us to assign reagents arrival times while keeping a low total force (~3.5 pN) on the DNA (Materials and methods) which is non-permissive for condensation by ParB. Furthermore, protein concentrations could be accurately controlled as the DNA was exposed within a few seconds (see below) to the condition of interest, allowing one to measure association and dissociation constants.

To characterise the switching between the two reagents, we first took advantage of magnetic tweezers to track a bead as it was driven by the flow alternation. We measured beads in both vertical (*Figure 3—figure supplement 1A*) and lateral (*Figure 3—figure supplement 1B*) configurations and demonstrated that changes in flow velocities correlate with the movement of the bead in the transverse direction. Importantly, the direction of the pulling force (x axis) was not disturbed when flow rates were switched (*Figure 3—figure supplement 1B*). Then, to directly monitor the boundary shifting, we measured fluorescence intensity changes in the background using a buffer containing 5 mM fluorescein at different flow rates (*Figure 3—figure supplement 1C and D*). Our results concluded that the boundary achieves a complete shifting in ~6.5 s.

We measured cycles of protein binding and unbinding by switching between channels containing ParB$^{AF}$ and buffer (*Figure 3B*, *Video 5*). The fluorescence intensity decreased when buffer was flowed (shadowed area) and it was recovered by the introduction of ParB$^{AF}$. Note that despite the limitation of fluid exchange kinetics, the binding curve of ParB$^{AF}$ on the DNA was readily different from that of the background. We performed experiments at different protein concentrations to elucidate the kinetics of ParB (*Figure 3—figure supplement 2*). The binding of ParB to DNA can be formulated as (*Goodrich and Kugel, 2007*):

$$\text{DNA} + \text{ParB} \underset{k_{off}}{\overset{k_{on}}{\rightleftharpoons}} \text{DNA:ParB} \tag{1}$$

with $k_{on}$ and $k_{off}$ being the binding and unbinding rate constants, respectively. This is a simple model which cannot account for cooperativity in binding nor potential nucleation intermediates, and hence it may not capture the detailed (un)binding kinetics of ParB. However, it is still useful for semi-quantitative analysis of binding rates as well as for direct comparison of the values at different conditions. Moreover, the dynamics of the system can be easily solved assuming that the concentration of ParB remains constant in time ($[ParB](t) = [ParB](0)$). This is a reasonable assumption, since in our single-molecule experiments we expect an excess of protein compared with DNA binding sites, and our measurement is much longer than the boundary shifting time. Then, the final expression for protein binding reads:

$$F(t) = F_{max}\left(1 - e^{-k_{obs}t}\right) \tag{2}$$

where $F(t)$ states for the fluorescence signal and the observed binding rate $k_{obs}$ is defined as by $k_{obs} = k_{on}\,[\text{ParB}] + k_{off}$. Similarly, the unbinding kinetics can be obtained by considering [ParB] = 0, and giving:

eLIFE Research advance                    Chromosomes and Gene Expression | Structural Biology and Molecular Biophysics

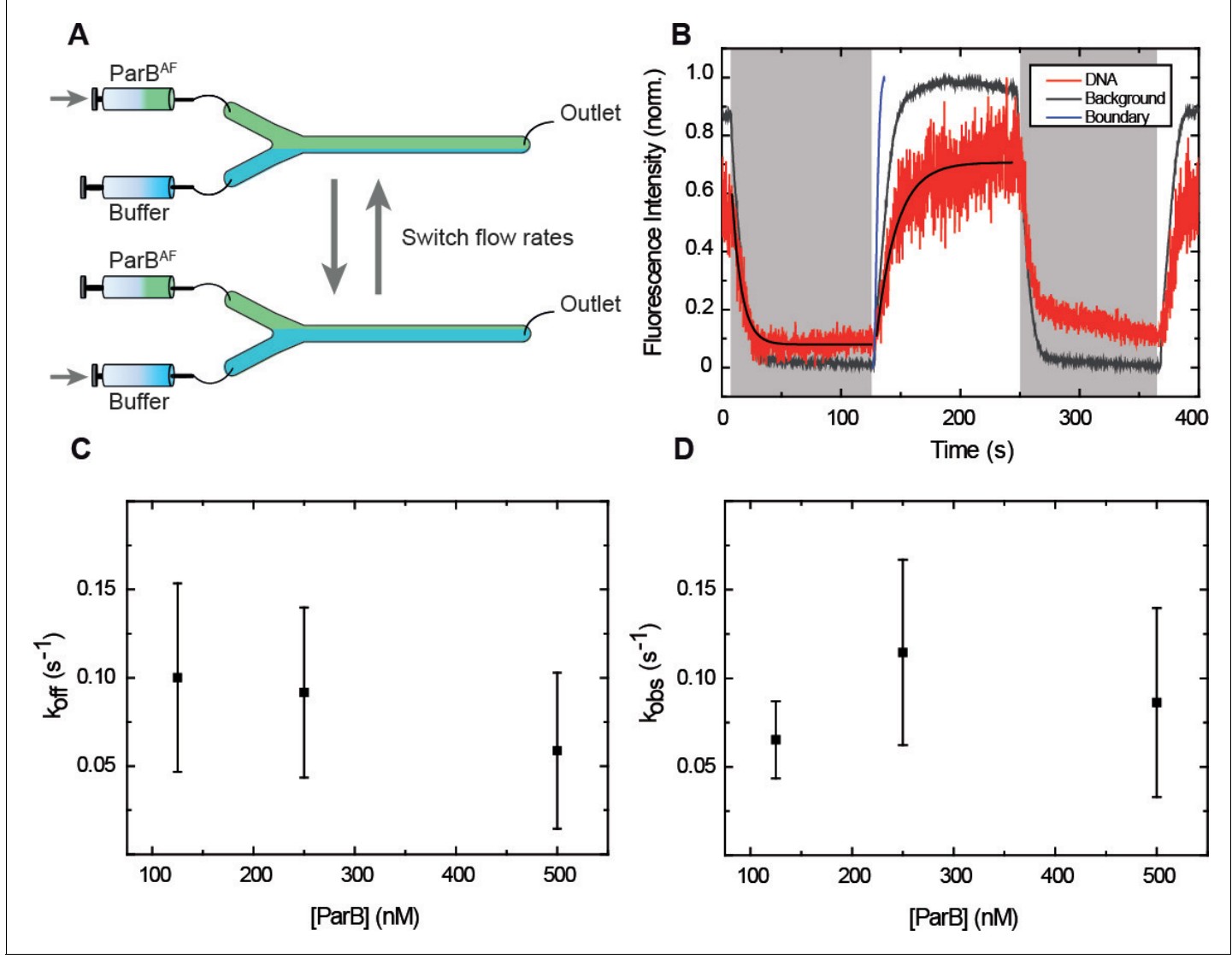

**Figure 3.** ParB binding and unbinding kinetics. (A) Scheme of the multilaminar flow system employed to fast-exchange of buffers. The fluid cell contains two inlets and a single outlet. Switching the velocities of both channels shifts the boundary of laminar flows resulting in the fast exchange of buffers. (B) Normalised integrated fluorescence intensity for a representative DNA molecule in a laminar-flow experiment with 250 nM ParB$^{AF}$ as a function of time (red) and for an equivalent background area (gray). Buffer injection (shadowed area) is correlated with decreasing intensity. Best fit curves to obtain $k_{off}$ (*Equation 3*) and $k_{obs}$ (*Equation 2*) are shown. Boundary exchange with fluorescein is also shown for comparative purposes (*Figure 3—figure supplement 1D*). (C) Unbinding rate $k_{off}$ measured at different initial ParB concentrations, obtained from fitting individual curves as shown in B, and then averaged at different concentrations. The values obtained are in the order of values published before (*Song et al., 2016*). (D) Observed binding rate $k_{obs}$ measured at different ParB concentrations, calculated as $k_{off}$. Errors are SD. (n ~ 15-30 molecules). All experiments were conducted in ParB reaction buffer supplemented with 4 mM Mg$^{2+}$.

DOI: https://doi.org/10.7554/eLife.43812.011

The following source data and figure supplements are available for figure 3:

**Source data 1.** Normalised integrated fluorescence intensity for a representative DNA molecule in a laminar-flow experiment with 250 nM ParB$^{AF}$ as a function of time.

DOI: https://doi.org/10.7554/eLife.43812.014

**Figure supplement 1.** Correct performance of the syringes demonstrated by tracking DNA tethers and boundary shift fluorescence measurements.

DOI: https://doi.org/10.7554/eLife.43812.012

**Figure supplement 2.** Measuring $k_{off}$ and $k_{obs}$ for different ParB$^{AF}$ concentrations.

DOI: https://doi.org/10.7554/eLife.43812.013

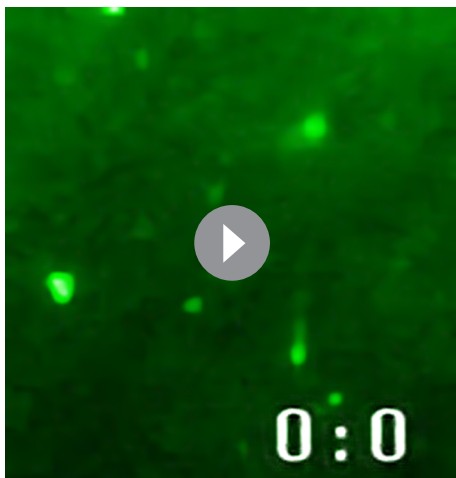

**Video 5.** Movie showing the laminar flow switching assay. 250 nM ParB$^{AF}$ binding increases intensity along a DNA molecule and unbinding decreases it (total flow rate 200 µl/min). Movie is 5X accelerated.
DOI: https://doi.org/10.7554/eLife.43812.015

$$F(t) = F_0\, e^{-k_{off}t} \qquad (3)$$

where $F_0$ is the initial fluorescence intensity (*Goodrich and Kugel, 2007*).

The measured binding constant $k_{off}$ (~0.1 s$^{-1}$) was independent of the protein concentration and slightly lower than the previously reported value for the condensation-deficient ParB mutant ParB$^{R82A}$, ~0.16 s$^{-1}$ (*Song et al., 2016*). This might reflect localised condensation events in the wild type protein (*Figure 3C*). This is particularly relevant at the high concentrations used (≥500 nM), where ParB-ParB interactions are favored likely resulting in local condensates, which prevent unbinding of the protein. Regarding $k_{obs}$ we observed a saturation effect at high concentrations (*Figure 3D*). This cannot be explained by limitations in the temporal resolution of our assay (acquisition rate ~10 s$^{-1}$) and we therefore favour the idea that it is limited by our fluid exchange kinetics. Nevertheless, the saturation observed can also be explained by rate limiting events after the formation of a collision complex (which are not considered in our simple model). We therefore considered $k_{obs}$ values obtained at low protein concentration to estimate $k_{on}$ (see Materials and methods). Our estimation (~4 x 10$^5$ M$^{-1}$ s$^{-1}$) was one order of magnitude below a published value for a mutant ParB, ~2.1 x 10$^6$ M$^{-1}$ s$^{-1}$ (*Song et al., 2016*) This discrepancy may reflect a different binding mechanism of the wild-type ParB protein employed here compared to the condensation-defective mutant, and/or simply the limitations of our laminar-flow-based technology.

## The C-terminal domain of ParB does not compete for DNA binding with full length ParB in Mg$^{2+}$ conditions

Motivated by previous work reporting a non-specific DNA binding of the free CTD of ParB and its important role in DNA condensation (*Fisher et al., 2017*), we investigated the effect of an excess of CTD in the ParB$^{AF}$ binding/unbinding reaction. Our previous work indicated distinct DNA-binding mechanisms for ParB in the presence of Mg$^{2+}$ or EDTA (*Taylor et al., 2015*). In particular, the specific binding of the protein was only observed in Mg$^{2+}$, while the non-specific binding was promoted by the presence of EDTA, especially at low ParB concentrations. In the work described below, we carried out assays in both Mg$^{2+}$ and EDTA conditions and measured $k_{off}$ and $k_{obs}$ for full length ParB, and for ParB in the presence of free CTD.

For full length ParB, protein:DNA complexes were observed under both conditions (*Figure 2B* and *Figure 4—figure supplement 1A*), but we measured a lower $k_{off}$ in EDTA conditions compared to Mg$^{2+}$ conditions (*Figure 4A*, *Figure 4—figure supplement 2*, *Table 1*). The observed binding rate ($k_{obs}$) did not show statistically significant differences between EDTA and Mg$^{2+}$ (*Figure 4A*, *Table 2*). To result in similar observed binding rates and considering the definition of $k_{obs}$, we infer that the association rate constant $k_{on}$ is higher in the presence of EDTA than in Mg$^{2+}$. In summary, and in agreement with our published observations (*Fisher et al., 2017*), we conclude that the presence of EDTA strengthens interactions between ParB and non-specific DNA.

We next produced a fluorescently labeled CTD protein (CTD$^{AF}$) and studied its binding to DNA in the presence of EDTA or Mg$^{2+}$ (*Figure 2—figure supplement 1*). We observed protein:DNA complexes at high CTD$^{AF}$ concentrations (10 µM) in the presence of EDTA but no complexes were visible in the presence of a magnesium-containing buffer at the same concentration (*Figure 4—figure supplement 1A*). The fluorescence intensity of CTD$^{AF}$ complexes remained constant over time, suggesting continuous and rapid exchange between bound and free CTD proteins (*Figure 4—figure supplement 1B* and *Video 6*). Further experiments, where CTD$^{AF}$ complexes were photobleached,

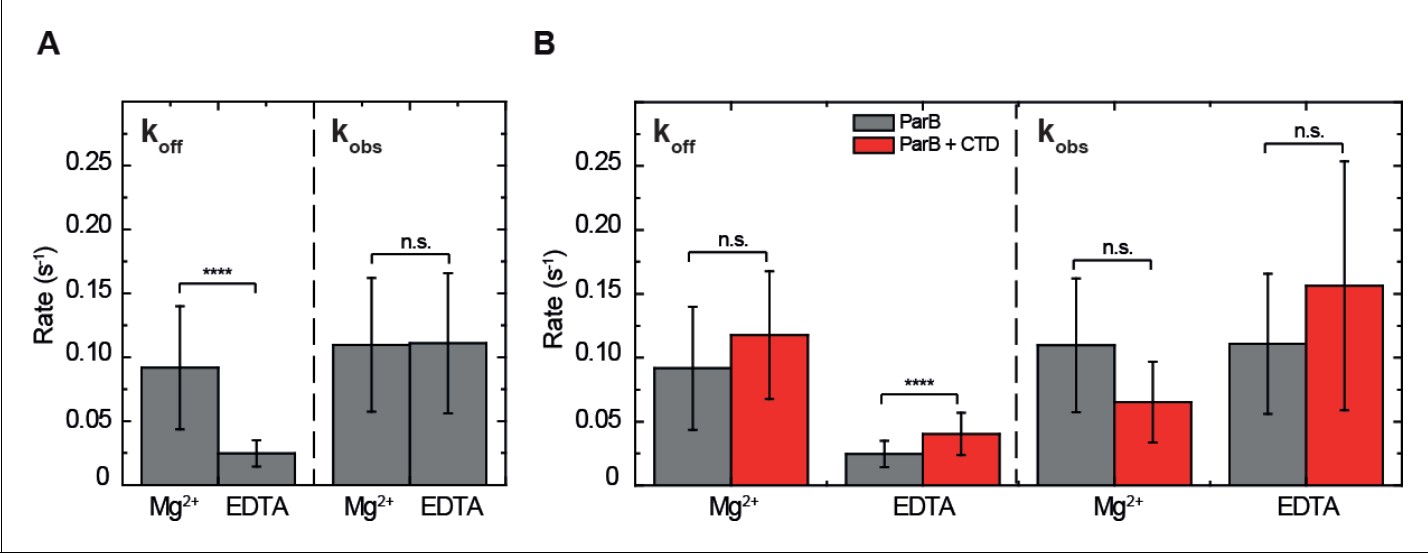

**Figure 4.** ParB binding kinetics in the presence and absence of $Mg^{2+}$ and the effect of the free CTD. (**A**) Unbinding rate $k_{off}$ and observed binding rate $k_{obs}$ values for 250 nM ParB[AF]. We observed a reduction in $k_{off}$ (slower unbinding) in the case of EDTA but no significant difference was observed in $k_{obs}$ values. (**B**) Unbinding rate $k_{off}$ and observed binding rate $k_{obs}$ values for 250 nM ParB[AF] in the presence (red) and absence (gray) of 2.5 µM CTD, in $Mg^{2+}$- or EDTA–containing buffer. We report faster unbinding in the presence of the CTD in EDTA. No effect of the CTD in the unbinding rate was observed in $Mg^{2+}$ buffer, according to Student's t-test. No significant difference was observed in $k_{obs}$ values in the presence or absence of the CTD in both $Mg^{2+}$ and EDTA conditions. All errors are SD. (n ~ 12-28 molecules). Data were accounted for statistical differences following a Student's t-test (see *Table 1* for $k_{off}$ p-values and *Table 2* for $k_{obs}$ p-values).
DOI: https://doi.org/10.7554/eLife.43812.016

The following figure supplements are available for figure 4:

**Figure supplement 1.** The CTD is capable of binding DNA.
DOI: https://doi.org/10.7554/eLife.43812.017

**Figure supplement 2.** Measuring the effect of the CTD on the dissociation rate in EDTA and $Mg^{2+}$ buffer conditions.
DOI: https://doi.org/10.7554/eLife.43812.018

confirmed the rapid exchange of the protein in a similar manner to ParB[AF] (see *Figure 4—figure supplement 1C and D* for data and discussion). Altogether, these results show that the CTD of ParB can bind and unbind rapidly from DNA, but only under EDTA conditions. This implies that the free CTD will not inhibit the interactions of full length ParB with DNA by simple competition under $Mg^{2+}$ conditions. We next tested this more directly.

In the presence of excess free CTD and under $Mg^{2+}$ conditions, the $k_{off}$ and $k_{obs}$ values for full length ParB were indeed unchanged when compared to full length ParB alone (*Figure 4B*). An excess of the CTD in EDTA-containing buffer facilitated ParB[AF] unbinding, yielded a higher $k_{off}$ (*Figure 4B*, *Figure 4—figure supplement 2*), and resulted in lower intensity CTD[AF]-DNA complexes. Therefore, as expected based on the properties of the free CTD alone, it can compete for DNA but only under EDTA conditions, which are not considered to have any particular biological relevance.

**Table 1.** Student's t-test p-values for $k_{off}$ values.

|  | WT - $Mg^{2+}$ | WT + CTD $Mg^{2+}$ | WT - EDTA | WT + CTD - EDTA |
|---|---|---|---|---|
| WT – $Mg^{2+}$ | 1 | 0.1235 | 5E-4 | 0.0019 |
| WT + CTD $Mg^{2+}$ |  | 1 | 0 | 0 |
| WT - EDTA |  |  | 1 | 6E-5 |
| WT + CTD - EDTA |  |  |  | 1 |

DOI: https://doi.org/10.7554/eLife.43812.019

**Table 2.** Student's t-test p-values for $k_{obs}$ values.

|  | WT - Mg$^{2+}$ | WT + CTD Mg$^{2+}$ | WT - EDTA | WT + CTD - EDTA |
| --- | --- | --- | --- | --- |
| WT - Mg$^{2+}$ | 1 | 0.079 | 0.9542 | 0.1139 |
| WT + CTD Mg$^{2+}$ |  | 1 | 0.0243 | 0.0103 |
| WT - EDTA |  |  | 1 | 0.1129 |
| WT + CTD - EDTA |  |  |  | 1 |

DOI: https://doi.org/10.7554/eLife.43812.020

These results strongly suggest that scenarios 1 and 2 (*Figure 1*) can be discounted, because the CTD does not reduce DNA binding by full length ParB under Mg$^{2+}$ conditions. However, these experiments were performed at restrictive forces to uncouple binding and condensation. At permissive forces, the Mg$^{2+}$ conditions would support condensation. We next investigated the dominant negative effect of the free CTD under conditions that were permissive for condensation.

### The free CTD prevents ParB-dependent condensation at permissive forces without affecting its binding to DNA

Previous experiments showed that an excess of the CTD of ParB prevents the condensation of DNA tethers and also promotes decondensation of preformed ParB-DNA networks (*Fisher et al., 2017*). However, as mentioned, we could not unequivocally distinguish between the possible mechanisms for condensation prevention, because binding and condensation could not be uncoupled. Here, using our MT-TIRF setup we could apply changes in force in a controlled way while at the same time visualising ParB binding and condensation.

We first monitored the condensation activity of ParB$^{AF}$ in a buffer supplemented with Mg$^{2+}$. Condensation events were initiated by ParB$^{AF}$ by reducing the force from 1 to 0.2 pN and were directly visualised by fluorescence microscopy in laterally stretched DNA molecules (*Figure 5A*, *Video 7*). This process could be stopped by increasing the force (see *Video 8*) but could not be reversed, possibly due to a strong interaction of the protein with the surface and the smaller range of available forces applied by the lateral magnet in comparison to previous conventional MT experiments (*Taylor et al., 2015*). In general, the condensation process did not show any brighter features along the DNA that could be associated with the formation of large loops or the presence of nucleation regions (*Figure 5A*, *Video 7*). Nevertheless, we should keep in mind that our setup has a limited resolution of about 1 kbp as determined from measurements on single fluorophores (data not shown). Therefore, local condensation at scales lower than this could not be resolved.

In vivo studies have shown that a very small number of ParB molecules are actually responsible for chromosome segregation in the bacterial cell (*Graham et al., 2014*). Inside the cell, the entire chromosome and all ParB dimers are of course compartmentalised leading to a situation in which [ParB] and [DNA-binding sites] are both well above the Kd values we observe for DNA binding, but are present in a stoichiometry such that there are many more DNA-binding sites than there are ParB dimers. Thus, ParB presumably ends up in foci around *parS* because of the high-affinity interaction between ParB and its specific site mediated by the HTH domain. This situation cannot possibly be achieved in vitro and in order to observe ParB:DNA binding we had to raise [ParB] to concentration over 100 nM. The minimum concentration tested that resulted in condensation was 250 nM ParB$^{AF}$.

We repeated condensation experiments using 250 nM ParB$^{AF}$ in the presence of 2.5 µM CTD in

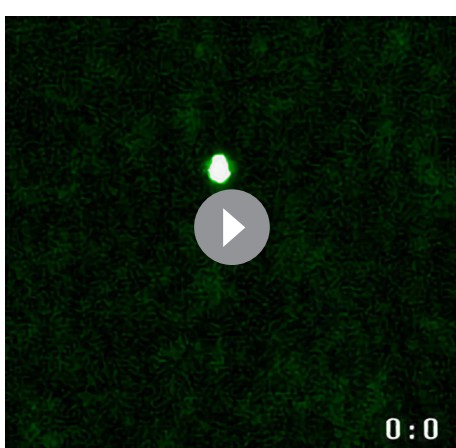

**Video 6.** Movie showing that intensity remains constant when imaging 10 µM CTD$^{AF}$, similar to ParB$^{AF}$. Movie is 5X accelerated.

DOI: https://doi.org/10.7554/eLife.43812.021

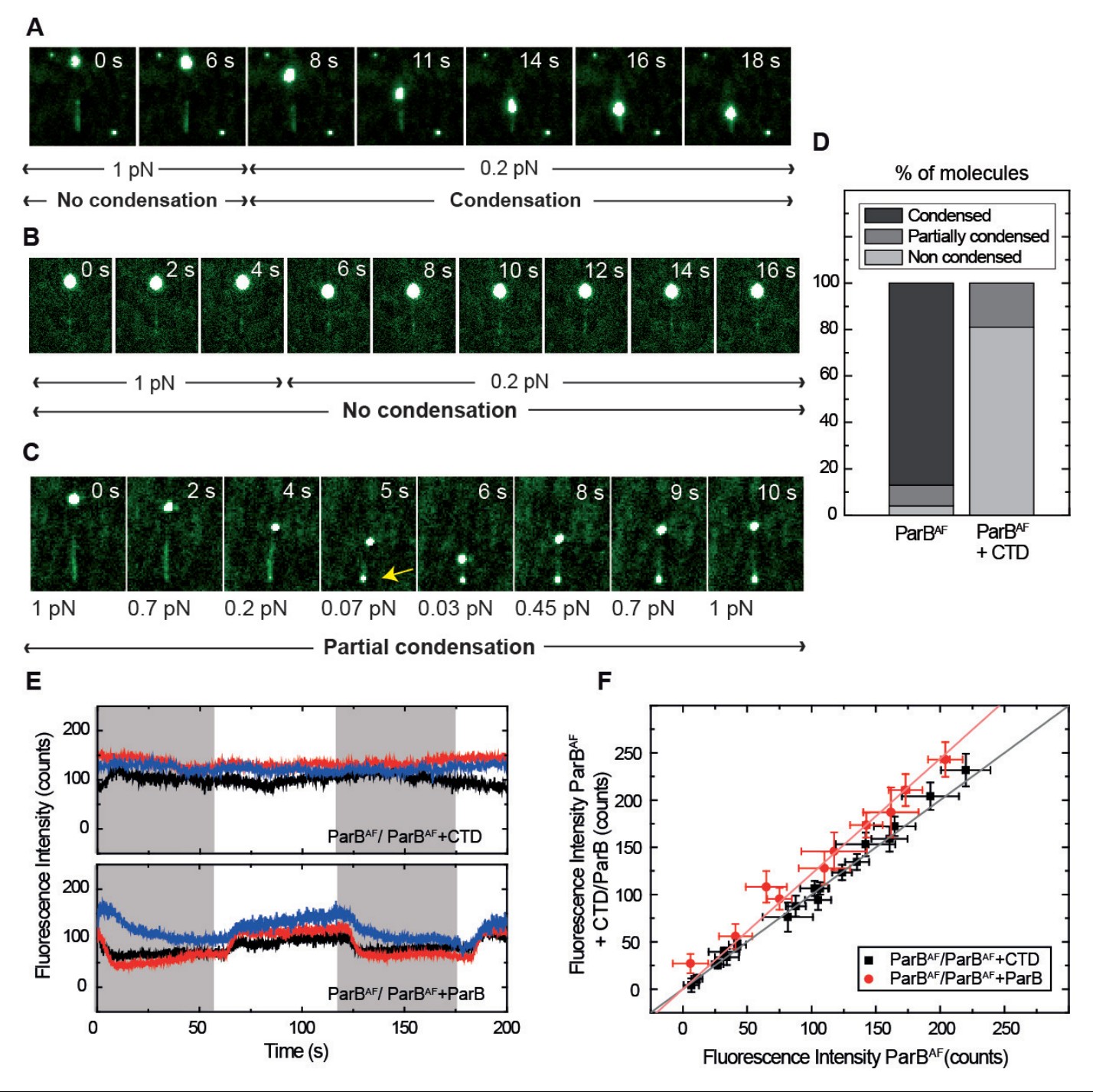

**Figure 5.** Competition of the CTD in ParB binding and unbinding kinetics, and effects in condensation. (**A**) Visualisation of condensation of a single DNA molecule induced by 250 nM ParB[AF] binding. The volume of the bead causes the DNA to be slightly tilted with respect to the surface such that emission of fluorescence is limited by the penetration depth of the excitation field. The DNA does not condense while held at non-permissive force of 1 pN. However, as in conventional magnetic tweezers experiments, at a permissive force of 0.2 pN a progressive reduction of the DNA length is observed. (**B**) Condensation experiment as in **Figure 5A** but in the presence of 2.5 μM non-fluorescent CTD showing that at a constant force of 0.2 pN, ParB is not able to condense to the inhibition of the CTD. Co-incubation with the CTD does not inhibit ParB binding to DNA, as indicated by clearly visible fluorescence filaments. (**C**) Condensation experiment as in **Figure 5B**. The CTD inhibits ParB-dependent DNA condensation, even at forces as low as 0.07 pN. Note that some residual condensation was observed at very low forces, visible at the anchoring point (arrow) as shown by the bright dot. (**D**) Percentage of DNA molecules that were condensed (more than 90% extension reduction), partially condensed or not condensed by ParB[AF] (less than 10% extension reduction), in the presence (n = 47) or absence of an excess of the CTD (n = 36). (**E**) Integrated fluorescence intensity in a

*Figure 5 continued on next page*

*Figure 5 continued*

laminar flow experiment. Intensity remains constant throughout the cycles of ParB$^{AF}$ and ParB$^{AF}$ + CTD (shadowed area), indicating that the excess of CTD is not competing for the binding sites on the DNA, but rather 'capping' the DNA-bound ParB$^{AF}$. Intensity changes when alternating ParB$^{AF}$ with a combination of ParB$^{AF}$ and unlabelled ParB (shadowed areas). (**F**) Integrated intensity of ParB$^{AF}$ plotted versus the integrated intensity of ParB$^{AF}$ + CTD or ParB$^{AF}$ + ParB. ParB$^{AF}$ + CTD shows a slope of 1.00 ± 0.02 corresponding to unchanged intensity, while ParB$^{AF}$ + ParB shows a slope of 1.23 ± 0.02. (n ~ 10–16 molecules). All experiments were conducted in ParB reaction buffer supplemented with 4 mM Mg$^{2+}$.
DOI: https://doi.org/10.7554/eLife.43812.022
The following source data is available for figure 5:

**Source data 1.** Integrated fluorescence intensity for 3 molecules throughout the cycles of ParB$^{AF}$ and ParB$^{AF}$ +CTD (constant intensity) or ParB$^{AF}$ and ParB$^{AF}$ +unlabelled ParB (intensity changes).
DOI: https://doi.org/10.7554/eLife.43812.023

Mg$^{2+}$ conditions (*Figure 5B*, *Video 9*). Only a very small fraction of DNA molecules showed some degree of condensation. Indeed, the time-lapse in *Figure 5C* (*Video 10*) shows that, even though the DNA molecule remains fully coated by ParB$^{AF}$, it did not condense, even at very low (permissive) forces. For the totality of molecules analyzed in this study, (47 in the case of ParB$^{AF}$ and 36 in the case of ParB$^{AF}$ + CTD), we found that 83% of the molecules were condensed by ParB$^{AF}$, whereas only 19% were condensed when the CTD was also present (*Figure 5D*). Moreover, when condensation was observed in the presence of the CTD, it was never complete. Typically, the extension of the molecule was not recovered upon re-application of high force, and a bright dot appeared at the anchoring point. The fact that normal condensation data (i.e. those collected with full length wildtype ParB only) do not show any such condensation foci (*Figure 5A*) and that the brighter dot in the CTD experiment appears at the anchoring point, suggests that this represents non-specific binding of the DNA to the surface that cannot be prevented at the low applied forces. The same bright dot was sometimes observed in the case of ParB$^{AF}$ at forces non-permissive for condensation, confirming our hypothesis (*Video 11*). We conclude that the presence of free CTD has converted the full length ParB into a form that can still bind, but can no longer condense DNA, which can only be explained by the capping model (*Figure 1D*).

We performed additional experiments using our laminar-flow system combined with direct visualisation of DNA-protein complexes. We switched between a buffer containing 250 nM ParB$^{AF}$ and another containing 250 nM ParB$^{AF}$ and 2.5 µM CTD in Mg$^{2+}$ conditions, whilst keeping the DNA stretched with lateral MT. The period of buffer exchange (T=60 s) is long enough to account for protein exchange since we measured a $k_{off}$ for ParB$^{AF}$ of 0.08 s$^{-1}$, corresponding to an occupation halftime of around 9 s (*Figure 3C*). Importantly, the intensity did not decrease when the conditions were switched from ParB alone to ParB co-incubated with CTD (*Figure 5E*, upper panel), again confirming a lack of competition for the DNA binding sites. Additionally, a control experiment where we substituted the excess of CTD for an equivalent excess of unlabelled ParB displayed a noticeable decrease in intensity attributed to the expected competition for DNA-binding sites between non-fluorescent ParB and ParB$^{AF}$ (*Figure 5E*, lower panel). To quantify this effect, we represented the average integrated intensity of ParB$^{AF}$ against the average integrated intensity of ParB$^{AF}$ + CTD/ParB for several molecules (*Figure 5F*). A deviation from a slope ~1 indicates changes in fluorescence intensity as a result of the competition for DNA-binding sites, validating the capability of our assay to qualitatively resolve differences in protein binding. Together, these results allowed us to conclude that condensation is impeded (or decondensation promoted) by the

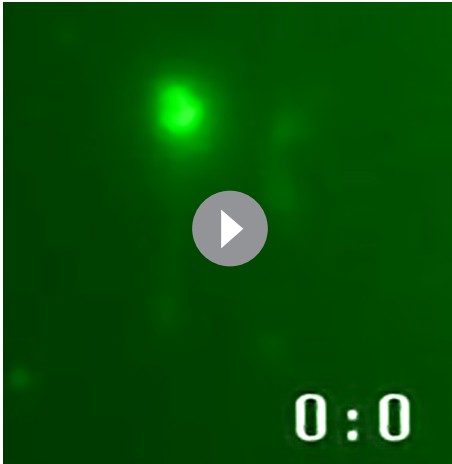

**Video 7.** Movie showing that 250 nM ParB$^{AF}$ is able to condense DNA under permissive forces. Movie is 5X accelerated.
DOI: https://doi.org/10.7554/eLife.43812.024

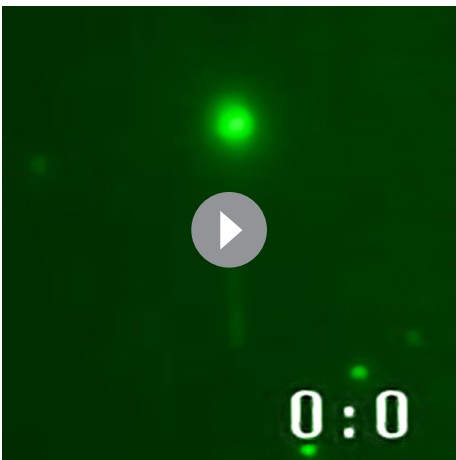

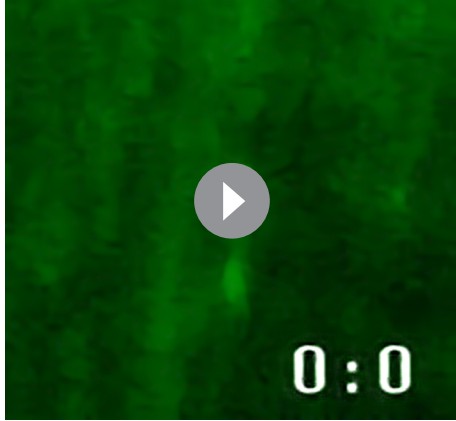

**Video 8.** Movie showing that 250 nM ParB$^{AF}$ does not decondense upon force increase, contrary to ParB in previously published magnetic tweezers assays. Movie is 5X accelerated.
DOI: https://doi.org/10.7554/eLife.43812.025

**Video 10.** Movie showing that 250 nM ParB$^{AF}$ in the presence of an excess of CTD (2.5 μM) is not able to condense DNA even in the absence of magnetic force. Movie is 5X accelerated.
DOI: https://doi.org/10.7554/eLife.43812.027

CTD as a result of capping a ParB-ParB dimerisation interface (*Figure 1D*).

## Conclusions

Here, we have built upon our previous work to incorporate MT and TIRF microscopy to visualise the formation of ParB-DNA complex structures at controlled forces that can be either permissive or non-permissive for condensation. Our novel approach allowed us to visualise the dynamic nature of the ParB binding with evidence for fast and continuous exchange of proteins with the surrounding media. This dynamic behavior is consistent with results described not only for *Bs*ParB (*Graham et al., 2014*) (*Taylor et al., 2015*) (*Song et al., 2017*) (*Fisher et al., 2017*), but also for plasmid ParB (*Debaugny et al., 2018*; *Sanchez et al., 2015*) (*Le Gall et al., 2016*), and for other proteins, at the single molecule level (*Gibb et al., 2014*). Implementation of a multilaminar flow exchange system to the combined MT-TIRF setup provided a measurement of binding and unbinding reactions in the presence of the free CTD leading to the conclusion that it only affects binding and unbinding of ParB in a buffer supplemented with EDTA. Fluorescence experiments with the CTD in a buffer supplemented with Mg$^{2+}$ strongly supported a mechanism for its dominant negative effect involving decondensation by direct disruption of ParB-ParB interfaces. Interestingly, the fact that the CTD prevents condensation by capping a dimerisation interface strongly suggests that the CTD in the full length ParB in solution is somehow inaccessible, unlike the DNA-bound ParB. This could explain why multimeric forms of ParB are not found in solution. The mechanism by which the CTD becomes accessible upon DNA binding is unclear and will be the subject of future research.

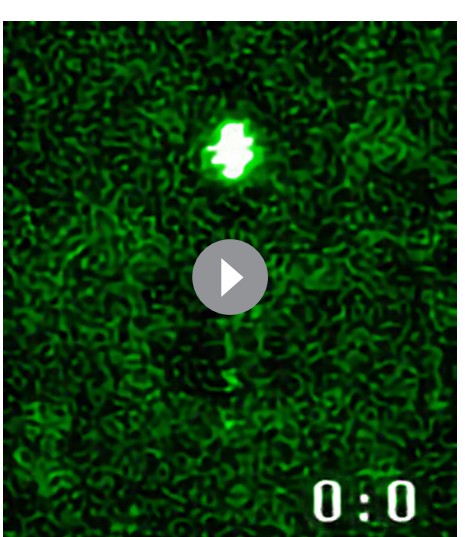

**Video 9.** Movie showing that 250 nM ParB$^{AF}$ in the presence of an excess of CTD (2.5 μM) is not able to condense DNA. Movie is 5X accelerated.
DOI: https://doi.org/10.7554/eLife.43812.026

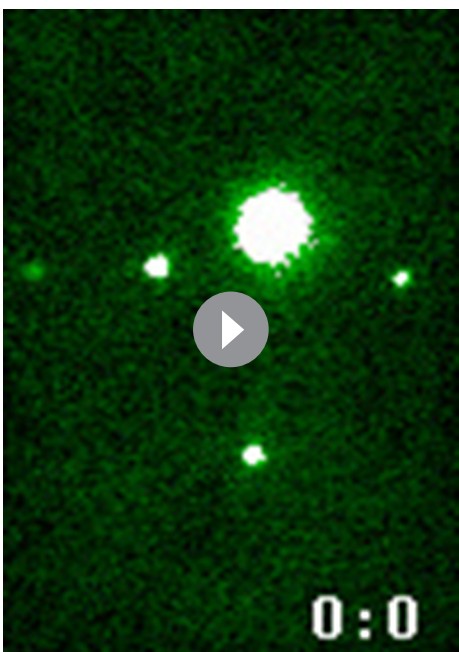

**Video 11.** Movie showing that a bright dot sometimes appears at the anchoring point of a DNA molecule in the presence of 250 nM ParB$^{AF}$. Movie is 5X accelerated.

DOI: https://doi.org/10.7554/eLife.43812.028

# Materials and methods

## Combined lateral MT-TIRF apparatus

Experiments were performed in two MT-TIRF set-ups similar to one described before (*Madariaga-Marcos et al., 2018*). In brief, 488 nm laser source (Vortran Stradus) was focused on the back focal plane of a high numerical aperture objective (Olympus UAPON TIRF 100×). We used two separate detectors to visualise the emission of the fluorophores in the sample and the magnetic beads; an EM-CCD temperature-controlled camera (Andor Ixon Ultra 897/Andor Ixon Ultra 888) and a CCD or CMOS camera (Pulnix 6710 CL/ Mikrotron MC1362) for bright-field video microscopy. Lateral Magnetic Tweezers consisted of a pair of permanent magnets (Q-05-05-02-G, Supermagnete) connected to a linear motor (Piezomotor). The MT setup was controlled by a custom-written code in *Umbarger et al. (2011)*. Fluorescence camera was controlled by Andor Solis software.

## Flow cells (simple and double channel)

In this work, two kinds of flow cells were used (with one or two inlets). In both cases, glass coverslips (Menzel-Gläser, #1) were cleaned by 30 min of sonication in acetone followed by 30 min in isopropanol, and dried using compressed air. The clean surface was coated with 1% polystyrene dissolved in toluene. The top cover glass contained two or three holes drilled with a laser engraver, as well as two-inlet paraffin wax (Parafilm M, Bernis USA) gaskets (VLS2.30, Universal Laser Systems). One-inlet gaskets were manually cut. The two cover glass slides and a gasket were sandwiched and heated up for a few seconds at 120°C to assemble the flow cell. The cells were then incubated with an Antidigoxigenin (25 ng/μl) solution (Roche) overnight at 4°C and were passivated for at least 2 hr using BSA (NEB). The cells were stored in a humid and sealed container at 4°C until further use.

## DNA substrate (λ/2)

λ/2 molecules were fabricated as described elsewhere (*Madariaga-Marcos et al., 2018*) (*Camunas-Soler et al., 2013*). Briefly, CosR-tail and CosL-tail oligonucleotides were biotin tailed and the XbaI-A oligonucleotide was digoxigenin tailed using Terminal Transferase (NEB) and either BIO-dUTP or DIG-dUTP (Roche). The modified oligonucleotides were purified using a Qiaquick nucleotide removal kit (Qiagen). N6-Mehtyladenine-free λ DNA (NEB) was cleaved with XbaI, giving two ~ 24 kbp fragments. These fragments and the tailed oligonucleotides in addition to the XbaI-B oligonucleotide were subsequently annealed and ligated overnight using T4 DNA ligase (NEB).

## Protein purification and labelling

*B. subtilis* ParB and the truncated CTD variant were recombinantly overexpressed in *E. coli* and subsequently purified using no extrinsic tags as described previously (*Taylor et al., 2015*) (*Fisher et al., 2017*).

ParB contains no native cysteine residues and so design of point substituted variants for covalent coupling to maleimide-conjugated fluorophores was pursued. Mutation S68C was selected because it is located in a surface-accessible putative loop region of poorly conserved sequence and length. ParB$^{S68C}$ was purified as described for the wild type protein with an additional labelling step.

Following elution from a Hiload 16/600 Superdex S75 gel filtration column (equilibrated in the absence of DTT), pooled concentrate was added to a 1 mg aliquot of Alexa Fluor 488 C5 Maleimide (Invitrogen) at a molar ratio of 1:5. This reaction was rotated end-over-end at 4°C for 16 hr before quenching by the addition of DTT to 5 mM. The conjugate solution was then loaded onto a Hiload 16/600 Superdex S75 gel filtration column (in the presence of 1 mM DTT), to separate labeled ParB (ParB$^{AF}$) and free fluorophore. ParB$^{AF}$ was pooled and concentrated before storage as described for wild type ParB. Apparent labelling efficiency was assessed by spectrophotometry as described by the vendor, by comparing the absorbance from the protein and the Alexa Fluor 488 dye, including a correction factor for absorbance of the dye at 280 nm (0.11). Native and collision-induced dissociation (CID) or electron-transfer dissociation (ETD) MS were used to assess labelling efficiency and stoichiometry, and to confirm covalent attachment of the fluorophore (not shown). ParB$^{AF}$ was fluorescently labelled at an approximate efficiency of 2 dyes per ParB monomer.

Similarly, the structure of the CTD dimer (PDB ID: 5NOC) was used to design a suitable site for cysteine substitution and fluorophore attachment. The residue S278 was selected as it is found in a relatively unconserved region located at the end of the final α-helix. Labelling reactions were performed as described before with the following modifications. Prior to labelling, the purified protein was treated with freshly prepared TCEP (at a final concentration of 1 mM), at room temperature for 1 hr with end-over-end rotation. A near stoichiometric molar ratio (~1:1.5) of the maleimide-conjugate dye was used over a 16 hr reaction course. Purification, quantification and storage of the protein-fluorophore conjugate were completed as described before (*Fisher et al., 2017*). CTD$^{AF}$ was labelled with a final efficiency of 0.82 dyes per CTD monomer.

## ParB/CTD binding and visualisation experiments

Tethers of λ/2 DNA molecules were obtained by mixing DNA with 1 µm size magnetic beads (Dynabeads, MyOne streptavidin, Invitrogen) in a buffer containing 100 mM NaCl, 50 mM Tris (pH 7.5), 0.2 mg/ml BSA, and 0.1% Tween 20, supplemented with 4 mM MgCl$_2$ (ParB reaction buffer) or supplemented with 1 mM EDTA (ParB-EDTA reaction buffer). DNA molecules were laterally stretched at non-permissive forces for condensation (over 1 pN). Then, 250 nM ParB$^{AF}$ or 10 µM CTD$^{AF}$ was injected into the cell at a flow rate of 250 µl/min and the DNA molecules were imaged using Andor Solis software. Images were acquired at a frequency of 9.52 Hz, using an EM level of 100 and cooling the sensor to −60 or −80°C. For condensation experiments, the lateral magnet was moved away from the flow cell to apply a force of 0.2 pN, while recording the fluorescence image. Fluorescence data analysis was performed in Andor Solis and Origin and movies were generated using ImageJ. For representation, images were exported from Andor Solis as 8-bit gray tiff files. First, a background was subtracted using a 50 pixel radius (sliding paraboloid, smoothing disabled) and then brightness and contrast were adjusted by visual inspection to enhance the signal. Finally, a custom look up table in green scale was applied. For the production of videos, fluorescence movies were exported as RGB single tiff files and assembled as videos using a custom ImageJ script (*Source code 1* provided). Both biological (new sample preparations from a fresh stock aliquot) and technical (single-molecule measurements) repeats were undertaken.

## Multichannel laminar-flow experiments for kinetics measurements

Tethers of λ/2 DNA were obtained by mixing DNA with 1 µm sized superparamagnetic beads in ParB or ParB-EDTA buffer supplemented with 1 mM Trolox, 20 mM glucose, 8 µg/ml glucose oxidase and 20 µg/ml catalase. The mixture was incubated and then flowed into the flow cell. After tethers were formed, unbound beads were extensively washed away. Tethers were laterally stretched at a force of 1 pN using lateral magnets. Fluorescence images were acquired using Andor Solis software. Images were acquired at a frequency of 9.52 Hz, using the EM level of 100 and cooling the sensor to −60 °C. 4000 frames were recorded. Syringes were controlled with the neMESYS UserInterface software. Briefly, a square-wave pattern was set for syringes to alternate flow-rates between 10 and 190 µl/min, keeping the flow-rate in the central channel to 200 µl/min. Employed ParB$^{AF}$ concentrations were 125, 250 and 500 nM and CTD$^{AF}$ was 10 µM. Data analysis was performed using Andor Solis and Origin. For each molecule of interest, a main region of interest (ROI) was selected around the DNA molecule in Andor Solis. Four additional ROIs around the main ROI were also selected for background correction by subtracting the mean intensity of the four ROIs to

the DNA ROI (in counts). Intensities were then analyzed in Origin. $k_{on}$ was estimated by fitting the first two points (corresponding to 125 nM and 250 nM) on **Figure 3D** to the equation $k_{obs} = k_{on} [\mathrm{ParB}] + k_{off}$, which yielded a value of ~3.95 x 10$^5$ M$^{-1}$ s$^{-1}$.

## Maximum Reynolds number in our microfluidic device

The Reynolds number is calculated as:

$$\mathrm{Re} = \frac{2\mathrm{r}\rho\,\mathrm{v_{mean}}}{\eta} \tag{4}$$

where $v_{mean}$ is the average linear velocity at the applied flow ($v_{mean} \approx 10^{-3}$ m s$^{-1}$ for the maximum flow rate of 200 µl/min), $\eta$ the dynamic viscosity of the fluid (10$^{-3}$ Pa s) and $\rho$ the density of the fluid (1 g cm$^{-3}$). We can determine the equivalent hydrodynamic radius of a square duct as $r = (d \cdot w)/(d + w)$, where $d$ and $w$ are the height and the width of the cross section of the liquid cell, respectively. In this case, $d \approx 200$ µm and $w \approx 3$ mm and we obtain a radius $r \approx 188$ µm. This gives a Reynolds number of $\mathrm{Re} \approx 0.001 \ll 2000$. Consequently, our system is always under laminar flow conditions.

## Calculation of the total force exerted on the DNA molecule while flowing under lateral magnetic pulling

The total force applied to the DNA is the sum of two components, the magnetic force exerted by the magnets and the drag force applied by the flow (**Madariaga-Marcos et al., 2018**):

$$\mathrm{F_{total}} = \sqrt{\mathrm{F_{magnetic}^2 + F_{drag}^2}} \tag{5}$$

The experiments reported here were done at a 1 pN pulling force, $\mathrm{F_{magnetic}} \approx 1$ pN. The drag force was deduced from the flow rate using a previously measured curve of force versus flow rate, in flow stretch assays $\mathrm{F_{drag}} \approx 3.36$ pN. This led to a total force $\mathrm{F_{total}} \approx 3.5$ pN.

## Fluorescence recovery after photobleaching (FRAP) experiments

We laterally stretched DNA tethers at a force of 1 pN. We then flowed 250 nM (or 1 µM for comparison with CTD$^{AF}$) ParB$^{AF}$. The imaging protocol consisted of imaging a DNA molecule at a low laser power for 200 frames before increasing the laser power for 100 frames to photobleach all ParB$^{AF}$ molecules. Fluorescence recovery was then allowed for 1200 frames at the initial laser power. Images were acquired at a frequency of 9.52 Hz, with an EM gain of 100 and with the sensor cooled to $-60$ or $-80$°C. The area (ROI) around each DNA molecule was analyzed using Andor Solis. Integrated fluorescence intensities were analysed using Origin.

## Acknowledgements

We are thankful for financial support from AEI/FEDER, UE (BFU2017-83794-P) and from European Research Council (ERC) under the European Union's Horizon2020 Research and Innovation programme (grant agreement no. 681299). JM-M acknowledges a predoctoral PhD fellowship from the Basque Country Government Department of Education, Language Policy and Culture (ref. PRE_2013_11_1174). GLMF was supported by the Biotechnology and Biological Sciences Research Council (1363883). MSD was supported by the Wellcome Trust (100401 and 077368).

## Additional information

### Funding

| Funder | Grant reference number | Author |
| --- | --- | --- |
| Basque Government | PRE_2013_11_1174 | Julene Madariaga-Marcos |
| Biotechnology and Biological Sciences Research Council | 1363883 | Gemma LM Fisher |
| Wellcome | 100401 | Mark Simon Dillingham |

| Wellcome | 077368 | Mark Simon Dillingham |
| Agencia Estatal de Investigación | BFU2017-83794-P | Fernando Moreno-Herrero |
| H2020 European Research Council | 681299 | Fernando Moreno-Herrero |

The funders had no role in study design, data collection and interpretation, or the decision to submit the work for publication.

## Author contributions
Julene Madariaga-Marcos, Conceptualization, Data curation, Formal analysis, Investigation, Methodology, Writing—original draft, Writing—review and editing; Cesar L Pastrana, Conceptualization, Data curation, Software, Formal analysis, Investigation, Methodology, Writing—review and editing; Gemma LM Fisher, Data curation, Formal analysis, Methodology, Writing—review and editing; Mark Simon Dillingham, Conceptualization, Supervision, Funding acquisition, Methodology, Writing—review and editing; Fernando Moreno-Herrero, Conceptualization, Supervision, Funding acquisition, Methodology, Writing—original draft, Project administration, Writing—review and editing

## Author ORCIDs
Julene Madariaga-Marcos (iD) http://orcid.org/0000-0003-1458-3369
Mark Simon Dillingham (iD) https://orcid.org/0000-0002-4612-7141
Fernando Moreno-Herrero (iD) http://orcid.org/0000-0003-4083-1709

## Decision letter and Author response
Decision letter https://doi.org/10.7554/eLife.43812.040
Author response https://doi.org/10.7554/eLife.43812.041

# Additional files

## Supplementary files
• Source code 1. Custom-written ImageJ script to assemble movies including a time stamp based on individual frames.
DOI: https://doi.org/10.7554/eLife.43812.029
• Transparent reporting form
DOI: https://doi.org/10.7554/eLife.43812.030

## Data availability
Data generated or analysed during this study are included in the manuscript and supporting files. Representative source data files are included as Videos. Analysis procedures are included in Materials and methods.

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
