## [Decision Letter]

Thank you for submitting your article "ParB dynamics and the critical role of the CTD in DNA condensation unveiled by combined force-fluorescence measurements" for consideration by *eLife*. Your article has been reviewed by three peer reviewers, one of whom is a member of our Board of Reviewing Editors, and the evaluation has been overseen by John Kuriyan as the Senior Editor. The reviewers have opted to remain anonymous.

The reviewers have discussed the reviews with one another and the Reviewing Editor has drafted this decision to help you prepare a revised submission.

Summary:

This manuscript, submitted here as a Research Advance, follows up on a previous study published in *eLife* last year demonstrating that the *Bacillus subtilis* ParB C-terminal domain (CTD) binds DNA non-specifically in addition to dimerization, and that this domain was required for DNA condensation. One of the key experiments showed a loss of ParB-induced condensation upon addition of free CTD domain. However, the exact mechanism underlying this process was not clear, as DNA binding could not be uncoupled from ParB-induced DNA condensation. In this current manuscript, this obstacle is overcome by simultaneously and independently probing ParB DNA binding and condensation using a combination of lateral magnetic tweezers and TIRF microscopy. These experiments clearly demonstrate that the CTD of ParB specifically inhibits its condensation activity without substantially affecting DNA binding when added in trans to full-length ParB. This leads the authors to the conclusion that the CTD interferes with ParB condensation through 'capping' of CTDs within ParB complexes as both the NTD and CTD dimerisation interfaces are needed for network formation.

Therefore, the manuscript represents a valuable addition and conclusion to the previous *eLife* paper, "The structural basis for dynamic DNA binding and bridging interactions which condense the bacterial centromere" by Fisher et al., (2017). The experiments are well-designed and overall, the results are presented clearly and support the authors' main conclusions. Nevertheless, a number of important points need to be addressed by the authors before the manuscript could be published.

Essential revisions:

1) The measurements of binding and unbinding kinetics are not consistent/convincing for a number of reasons. First, since k_obs_ = k_off_ + k_on_, it seems strange that in a number of cases (e.g., in Figure 3C and D, as well as 4A with Mg), k_off_ is either indistinguishable or higher than k_obs_. Based on the data from Fisher et al., 2017, Kd for ParB non-specific DNA binding is on the order of few hundred nM, and therefore in the tested concentration range k_on_ and k_off_ are expected to be comparable. Also, the fact that the equilibrium fluorescence level of ParB (Figure 3—figure supplement 2) changes with ParB concentration means that (assuming that k_off_ is independent of ParB concentration) k_on_ has to change at least 3-fold, whereas it doesn't seem to significantly change based on Figure 3C. Combined with the fact that the estimated k_on_ is an order of magnitude lower than a previously published result for a ParB mutant (Song et al., 2016), this raises the question of how accurate the kinetic measurements presented in this manuscript might be. One possible explanation would be that the measurement is dominated by liquid exchange kinetics. The authors should provide data on the concentration of ParB in solution during buffer exchange (from the background intensity). An independent measurement of ParB-DNA binding kinetics using a different method would be needed to support the TIRF k_obs_ and k_off_ measurements. Generally, the statistics for these experiments are insufficient and the authors need to improve them and address above issues. That being said, the kinetic measurements are not critical for the main conclusion of the paper.

2) Previously, the authors have demonstrated that CTD forms a highly stable dimer in solution with a characteristic exchange time of hours to days (Fisher et al., 2017), but the proposed capping mechanism of ParB condensation inhibition involves a CTD monomer binding to full-length ParB. The authors should demonstrate the CTD monomer exchange that seems to be necessary for the proposed model. One way to do this would be to use a CTD labeled with a quencher or a FRET acceptor for A488 (e.g., Cy3). In that case, the formation of heterodimers with A488-CTD and their exchange in the presence of unlabeled partners could be monitored using bulk fluorescence (or even TIRF microscopy). Otherwise, the authors need to provide a convincing explanation for this apparent contradiction.

3) The author's conclusion that the ParB CTD might be inaccessible in solution is appealing, as it echoed with the observation that *Bacillus subtilis* and *Thermus thermophilus* ParB both exist in solution in a monomer-dimer equilibrium while the CTD alone is a dimer (Leonard et al., 2006, Taylor et al., 2015). Additional experiments could be carried out by the authors to strengthen and develop their hypothesis, using hydrogen-deuterium exchange mass spectrometry (HDX) for example. Since the current manuscript is meant to be a single advance, this is best kept for a future, more in-depth study.

4) The authors have mentioned that the fluorescent labeling of proteins was controlled using mass spectrometry. These data need to be included in the paper, in particular in light of the puzzling reported labeling efficiency for ParB (2 dye molecules per monomer containing a single cysteine as stated in the captions for Figure 2A). The authors should discuss the possible reasons for a labeling efficiency that is substantially higher than 1.

5) An intriguing result presented by the authors is the lack of DNA-binding activity from the CTD in an Mg^2+^-supplemented buffer (Figure 4—figure supplement 1A), contrary to ParB in the same conditions. Although not addressed by the authors in their manuscript, this is highly reminiscent of their hypothesis (Fisher et al., 2017) of a second non-specific DNA binding site (possibly on the CDBD). In that case, this second site would still contribute to non-specific DNA binding in Mg^2+^ conditions while the CTD does not bind anymore, resulting in different binding kinetics which could be those measured in Figure 4A in Mg^2+^-supplemented buffer. This idea could be tested with labelled NTD-CDBD and/or labelled CDBD. One could also try to characterise the binding kinetics of the CTD in EDTA to compare it to ParB's.

6) The authors studied DNA condensation with 250 nM ParBAF. In vivo studies (Graham et al., 2014) have shown that a very small number of ParB molecules was actually responsible for chromosome segregation in the bacterial cell. In their setup, as the authors state, "an excess of protein [is expected] compared to DNA binding sites". Did the authors test what was the lowest concentration of ParB they could get to condense DNA? It would also be interesting to compare condensation between the λ/2 DNAs used in this study and DNAs harbouring one or several parS sites, but this is beyond the scope of this study.

7) It is not very clear if the experimental conditions for condensation inhibition are the same here than in the previous authors' published work. Indeed, previous experiments seem to have been performed in buffer without Mg^2+^ and without EDTA if I understand correctly ("100 mM NaCl, 50 mM Tris-HCl or HEPES-KOH pH 7.5, 100 mg/ml BSA and 0.1% Tween 20 (v/v)" according to Materials and methods section from Fisher et al., 2017). Please clarify.

---

## [Author Response]

Essential revisions:1) The measurements of binding and unbinding kinetics are not consistent/convincing for a number of reasons. First, since k_obs_ = k_off_ + k_on_, it seems strange that in a number of cases (e.g., in Figure 3C and D, as well as 4A with Mg), k_off_ is either indistinguishable or higher than k_obs_. Based on the data from Fisher et al., 2017, Kd for ParB non-specific DNA binding is on the order of few hundred nM, and therefore in the tested concentration range k_on_ and k_off_ are expected to be comparable. Also, the fact that the equilibrium fluorescence level of ParB (Figure 3—figure supplement 2) changes with ParB concentration means that (assuming that k_off_ is independent of ParB concentration) k_on_ has to change at least 3-fold, whereas it doesn't seem to significantly change based on Figure 3C. Combined with the fact that the estimated k_on_ is an order of magnitude lower than a previously published result for a ParB mutant (Song et al., 2016), this raises the question of how accurate the kinetic measurements presented in this manuscript might be. One possible explanation would be that the measurement is dominated by liquid exchange kinetics. The authors should provide data on the concentration of ParB in solution during buffer exchange (from the background intensity). An independent measurement of ParB-DNA binding kinetics using a different method would be needed to support the TIRF k_obs_ and k_off_ measurements. Generally, the statistics for these experiments are insufficient and the authors need to improve them and address above issues. That being said, the kinetic measurements are not critical for the main conclusion of the paper.

We thank the reviewers for raising this point regarding the accuracy of our kinetic measurements. Our kinetic measurements used a simple bimolecular association model as the framework for the analysis. If this model holds then k_off_ is expected to be independent of [ParB], whereas k_obs_ (the pseudo first order binding rate) should increase linearly with [ParB] above the value for k_off_. This is not the case and both rates appear to be limited to ~0.1 s^-1^.

This could be interpreted in a number of ways. On the one hand, this could indicate that the model we are using to describe the system is too simplistic (which is almost certainly the case given that DNA binding is apparently coupled to protein association). However, the reviewers are quite correct to highlight that another possible explanation for this behavior would be an artefactual rate-limiting event for both processes associated with fluid exchange. Indeed, we were concerned ourselves about this, and have performed additional experiments to determine the time of fluid exchange in our setup.

Although our Magnetic Tweezers measurements showed an apparent instantaneous boundary exchange (on the order of 500 ms, see Figure 3—figure supplement 1), further characterization of our experimental system shows that the effective fluid exchange rate is slower. We recorded the fluorescence at the surface under TIRF illumination to characterize the boundary switching between 5 mM fluorescein and buffer (Figure 3—figure supplement 1C-D) and found that it is achieved within 3-6.5 s (90% of saturating intensity), at 600-200 μl/min flow rates. We fit the Taylor-Aris model (Taylor, 1953) (Aris, 1956) for a rectangular channel as described in (Niman et al., 2013). The fit was very good as expected for a boundary fluid exchange, in contrast with the exponential rise expected for protein association. Additional experiments using 1 mM fluorescein and 100 nM SYTOX Green showed very similar results (not shown). These measurements were also found to be dependent on the quality of the surface, which varies between flow cells. Additionally, our measurements include effects from fluorophore adsorption and photobleaching which are difficult to account for. All of these factors contribute to the measured signal and highlight the difficulty in measuring the liquid exchange kinetics. However, we found that the kinetics of protein binding to DNA were readily different from those observed in the background, and in the fluorescein experiment. In the example shown in new Figure 3B, the k_off_ value for the DNA was ~0.09 s^-1^ whereas for the background it was ~0.12 s^-1^. Similarly, k_obs_ for the DNA was ~0.06 s^-1^ and ~0.09 s^-1^ for the background. These figures have been added to Figure 3—figure supplement 1 and the new information to the kinetics section of the text (subsection “Association and dissociation kinetics of ParB measured at non-permissive forces for condensation”).

Nevertheless, we concede that the limitation in the non-instantaneous reagent exchange may limit an absolute determination of the kinetic rates in our measurements as may also be the case for similar approaches based on DNA flow stretching and visualization by fluorescent dyes and/or proteins.

Considering that the binding and unbinding rate of ParB to and from DNA can be clearly differentiated from that of the background rate, our semi-quantitative measurements are still useful for comparison purposes. Moreover, the results of k_obs_/k_off_ in the presence of Mg^2+^/EDTA and in the presence or absence of the CTD (Figure 4) are compatible with the results obtained in bulk (Figure 2—figure supplement 1) and in equilibrium single-molecule experiments (Figure 2 and Figure 4—figure supplement 1). We have replaced Figure 3B by an updated version Figure that includes the background signal and modified the main text accordingly (subsection “Association and dissociation kinetics of ParB measured at non-permissive forces for condensation”).

Additionally, we have calculated the average intensity at equilibrium in kinetic binding experiments and have represented them as a function of protein concentration (new Figure 3—figure supplement 2B). On top of these data, we generated a theoretical Hill plot considering the K_d_ and Hill values obtained by bulk binding measurements in Taylor et al., 2015 and the two sets of data are clearly in reasonable agreement (see new Figure 3—figure supplement 2). Bulk stopped-flow measurements of binding kinetics were also reported in the Supplementary Information of Taylor et al. These were complex, but like the single molecule measurements very slow, with an observed halftime of ~10 s at 1 µM ParB. Note however, that these measurements are not comparable as the PIFE measurement likely measured the kinetics of both binding and condensation as opposed to our single molecule measurements which measure binding only under conditions non-permissive for condensation.

Finally, we appreciate the reviewers’ comment concerning the statistics. However, as pointed out also by the reviewers, the values of the rates determined from our semiquantitative kinetic measurements are not essential for the general message of the paper. Nevertheless, they are important qualitatively to support for instance that CTD is only capable of binding to DNA in the presence of EDTA (Figure 4B).

2) Previously, the authors have demonstrated that CTD forms a highly stable dimer in solution with a characteristic exchange time of hours to days (Fisher et al., 2017), but the proposed capping mechanism of ParB condensation inhibition involves a CTD monomer binding to full-length ParB. The authors should demonstrate the CTD monomer exchange that seems to be necessary for the proposed model. One way to do this would be to use a CTD labeled with a quencher or a FRET acceptor for A488 (e.g., Cy3). In that case, the formation of heterodimers with A488-CTD and their exchange in the presence of unlabeled partners could be monitored using bulk fluorescence (or even TIRF microscopy). Otherwise, the authors need to provide a convincing explanation for this apparent contradiction.

This question was already a topic of discussion for our last paper and the short answer is (somewhat surprisingly), yes, it is possible, and there is no apparent contradiction between the model and the data. Our NMR experiments showed previously that CTD monomers can exchange. However, the reviewers’ intuition says that, since the timescale of CTD exchange is on the orders of hours/days, this surely cannot be consistent with the relatively rapid exchange (on the order of seconds/minutes) proposed to occur in the MT experiments.Using kinetic simulations, we rigorously demonstrate below that it is possible to have rapid monomer exchange even though the dimer is extremely tight. This requires two special conditions to be satisfied: (A) one partner in the exchange must have an artificially elevated k_off_ and (B) the other partner must be at a relatively high concentration and large stoichiometric excess. Both of these conditions are satisfied in our experiment.

A) The ParB on the DNA experiences a restrictive force against condensation which, by definition, will increase the off rate for whatever molecular interaction bridges between ParB molecules to promote condensation (we propose that these include interactions between ParB CTDs). k_off_ increases exponentially with applied force (Schwesinger et al., 2000), such that relatively small force increases can result in very large effects on molecular interactions. Note that, the force in our experiment can easily reach a level (just a few piconewtons) at which the molecular interactions holding the condensate together are instantaneously broken (i.e. k_off_ is much faster than the experimental timescale).

B) The free CTD in solution is present at 5 µM, and in a huge stoichiometric excess compared to the ParB bound to DNA on the coverslip.

The kinetic simulations below use the same model to compare monomer:monomer exchange between tight dimers under four different scenarios.

1) The exchanging partners (dimers) are present at equimolar concentration in the μM range and there is no opposing force on either. The exchange is very slow as would be expected, as is observed in NMR experiments, and as the referees’ intuition suggests.

2) As 1, but one of the exchanging partners is present at a dramatically lower concentration. The exchange is still very slow.

3) As 1, but one of the exchanging partners experiences a restrictive force against the dimer interface thereby elevating its k_off_ by 1000-fold. Nevertheless, the exchange remains very slow because the system is rate limited by the slowly dissociating partner.

4) As 1, but one of the exchanging partners is present at a dramatically lower concentration AND one of the exchanging partners experiences a restrictive force against the dimer interface thereby elevating its k_off_ by 1000-fold. The exchange is now very fast.

This phenomenon can be rationalised by considering the fact that only a tiny fraction of the free CTD dimer must spontaneously dissociate in order to cap the full length ParB proteins bound to DNA, whose CTDs are frequently exposed due to the applied force. Thus, although the rate constant for CTD monomerization is very slow indeed (in units s^-1^), the absolute rate of monomer formation (nM s^-1^) is sufficiently high (given the high relatively concentration of the CTD dimer) to facilitate fast exchange and efficient capping.

For the particular case of scenario 4 (similar to our experimental conditions) we obtain a halftime of about 10 s in contrast with scenarios 1-3 that give values of 7000 s. Details of the kinetic model and simulations for all scenarios are included at the end of our responses under the heading Kinetic Model.

For some time, we also attempted to directly demonstrate experimentally the binding of the CTD monomers to ParB. We initially tried the experiment of injecting CTD^AF^ into a ParB-coated DNA molecule. However, the large concentrations of the CTD required caused a very high fluorescence background which prevented us from being able to observe any fluorescence that might have occurred on the ParB-coated DNA. The alternative idea suggested by the reviewer of using a quencher is appealing but these materials would have to be fabricated and properly tested before use. Instead, we have followed an alternative approach similar to the procedure described in the manuscript to determine the photobleaching rate of ParB^AF^ (Figure 2—figure supplement 2B). In brief, we co-incubated ParB (non-labeled) with a large excess of CTD^AF^ in Mg^2+^ conditions (which are restrictive for effective binding of CTD alone to DNA) together with 1% formaldehyde. Formaldehyde will crosslink DNA and proteins. After 15 minutes we washed the non-bound complexes by injecting a protein-free buffer. As expected based on our favored model, we observed fluorescent proteins (CTD^AF^) crosslinked to the DNA by the formaldehyde, which progressively photobleached (Author response image 1). Note that this method gives a very high signal-to-background ratio because it allows one to work with a minimal fluorescence background. This experiment shows that the CTD binds preferentially to DNA-bound ParB, under Mg^2+^ conditions (Author response image 1 right panel). This experiment was however imperfect, in that we observed some binding of CTD^AF^ alone to the DNA with formaldehyde (Author response image 1 left panel), albeit much lower than when a mixture of ParB and CTD^AF^ was used. These experiments support the CTD capping model where CTD dimers dissociate and bind to ParB DNA-bound proteins inducing decondensation (Scenario 3, Figure 1D).

**Author response image 1. respfig1:** CTD^AF^ binds to full-length ParB bound to DNA. (**A**) Fluorescence signal of CTD^AF^ crosslinked with formaldehyde in Mg^2+^ conditions in the absence of ParB (left) and in the presence of ParB (right). (**B**) Quantification of fluorescence signal in both experiments. This experiment supports the model of CTD^AF^ dimers dissociate to bind to ParB proteins bound to DNA (Scenario 3, **Figure 1D**).

3) The author's conclusion that the ParB CTD might be inaccessible in solution is appealing, as it echoed with the observation that Bacillus subtilis and Thermus thermophilus ParB both exist in solution in a monomer-dimer equilibrium while the CTD alone is a dimer (Leonard et al., 2006, Taylor et al., 2015). Additional experiments could be carried out by the authors to strengthen and develop their hypothesis, using hydrogen-deuterium exchange mass spectrometry (HDX) for example. Since the current manuscript is meant to be a single advance, this is best kept for a future, more in-depth study.

We agree with the reviewers’ point here. The idea that the CTD is somehow shielded from intermolecular interaction in the absence of DNA, and becomes available for interactions that lead to multimerization in the presence of DNA is indeed attractive; it can explain why a protein with two (perhaps three) protein:protein interfaces can exist as a dimer when alone in free solution but assemble into large networks in the presence of DNA. It is essentially the model presented by Leonard *et al.* in their structural paper (albeit in a slightly different context because we now have different thoughts about the nature of ParB networks). Dillingham’s PhD student is currently performing FPOP- and HDX-coupled MS analysis of ParB in the absence and presence of DNA. This is a collaboration with the Sobott group (Leeds) and the data will not be available and/or processed on a useful timescale for this re-submission.

4) The authors have mentioned that the fluorescent labeling of proteins was controlled using mass spectrometry. These data need to be included in the paper, in particular in light of the puzzling reported labeling efficiency for ParB (2 dye molecules per monomer containing a single cysteine as stated in the captions for Figure 2A). The authors should discuss the possible reasons for a labeling efficiency that is substantially higher than 1.

ParB^S68C^ was purified in the same manner as wild type ParB and labelled using a standard (vendor-supplied) protocol using a maleimide reagent. This yielded fully-functional (i.e. DNA binding proficient) ParB with an average of 2.4 dyes/monomer based on absorbance measurements, which was unexpected given that the protein contains a single engineered Cys (Author response image 2). Native mass spectrometry shows that the monomer contains between 1-4 labels (with 2 labels being by far the most abundant population) and that the dimer contains between 2 and 7 labels (with 4 labels being the most abundant) (Annika Butterer and Frank Sobott, personal communication). This is independent of whether the protein is folded or not and so these labels are covalently attached to the protein. Note that maleimides also react with amines, especially if they are activated by their local environment, and this is the most likely explanation for the observed labelling. This over-labelling is nevertheless unusual (e.g. our laboratory has labelled a very large number of different proteins using this chemistry and we have never come across this extent of non-specific labelling before). The over-labelling is reproducible and we also know that the wild type protein (with no cysteines) is efficiently labelled with ~ 1 dye/monomer, presumably reflecting the same phenomenon. However, regardless of this non-specific labelling activity, it is critical to note that all of our labelled full length ParB proteins retain wild type DNA binding activity.

Since our proteins display wild type activity, the over-labelling detected for ParB was not considered to be relevant.

**Author response image 2. respfig2:** Preparation of functional fluorescently labelled ParB. (**A**) SDS-PAGE of finally purified ParB^S68C^ and wild type ParB, selectively and non-selectively, respectively, conjugated to Alexa 488. (B-D) Representative TBM- and TBE-EMSAs to assess the activity of the fluorophore-labelled proteins. Wild-type-like specific and non-specific DNA-binding activity is retained. In-gel detection of the fluorescent protein corresponds to the pattern of nucleoprotein complexes.

5) An intriguing result presented by the authors is the lack of DNA-binding activity from the CTD in an Mg^2+^-supplemented buffer (Figure 4—figure supplement 1A), contrary to ParB in the same conditions. Although not addressed by the authors in their manuscript, this is highly reminiscent of their hypothesis (Fisher et al., 2017) of a second non-specific DNA binding site (possibly on the CDBD). In that case, this second site would still contribute to non-specific DNA binding in Mg^2+^ conditions while the CTD does not bind anymore, resulting in different binding kinetics which could be those measured in Figure 4A in Mg^2+^-supplemented buffer. This idea could be tested with labelled NTD-CDBD and/or labelled CDBD. One could also try to characterise the binding kinetics of the CTD in EDTA to compare it to ParB's.

Originally, a CTD deletion construct was designed based on secondary structure predictions, in which a stop codon was introduced following residue E222 (ParB^ΔCTD^ E222). Somewhat surprisingly, TBM- and TBE-EMSA analysis showed that this construct was defective in *parS* DNA binding, even when using concentrations 10-fold higher than required by wild type ParB. However, our structural model building later suggested that E222 resides within a structured α-helix that is part of the CDBD (helix 11). Modelling instead suggested that E227 may be a better truncation point for a clean removal of the CTD, leaving the NTD/CDBD domains intact (ParB^ΔCTD E227^). This position appears to be in the linker between the final α-helix of the CDBD and first β-sheet of the CTD. Expression trials confirmed that this construct was soluble, and for ease of purification, an N-terminal His-tag with a HRV 3C cleavage site, was used (Author response image 3). EMSA analysis showed that ParB^ΔCTD E227^, similar to ParB^ΔCTD E222^, had impaired *parS* and nonspecific DNA-binding activity (Figure Author response image 3). Given that the HTH motif seems to be the major structural determinant for *parS* binding, this result suggests it does not function and/or is not folded correctly, in the absence of the CTD. Clearly, the functional interplay of domain organisation of *B. subtilis* ParB is complex. We have never attempted to make a CDBD only construct, given the complication, with making a functional NTD-CDBD only construct.

Finally, we tried, as the referee suggests, to measure the binding kinetics of the CTD in EDTA conditions. However, it was not possible to obtain reliable (un)binding kinetic curves of CTD in EDTA due to the low signal-to-background ratio observed for these particular conditions that include high CTD concentrations (Figure 4—figure supplement 1A).

**Author response image 3. respfig3:** The CTD of ParB is required for the correct folding of the N-terminal and/or central DNA-binding domains. (**A**) SDS-PAGE of wild-type ParB and ParB^ΔCTD E227^. 2-6 µg of each was loaded as indicated. (B-C) Representative TBM- and TBE- EMSAs for ParB^ΔCTD E227^ monitoring binding of parS-containing or non-specific 147 bp DNA.

6) The authors studied DNA condensation with 250 nM ParBAF. In vivo studies (Graham et al., 2014) have shown that a very small number of ParB molecules was actually responsible for chromosome segregation in the bacterial cell. In their setup, as the authors state, "an excess of protein [is expected] compared to DNA binding sites". Did the authors test what was the lowest concentration of ParB they could get to condense DNA? It would also be interesting to compare condensation between the λ/2 DNAs used in this study and DNAs harbouring one or several parS sites, but this is beyond the scope of this study.

We have previously published an in-depth study of the relationship between ParB concentration, DNA binding, and DNA condensation (Taylor et al., 2015). We think the reviewers are implying that the condensation observed in vitro may not occur under more physiologically relevant conditions. This touches upon an important technical issue around the study of ParB in vitro which we acknowledge. Inside the cell, the entire chromosome and all ParB dimers are of course compartmentalized, leading to a situation in which [ParB] and [DNA binding sites] are both well above the Kd values we observe for DNA binding, but are present in a stoichiometry such that there are many more DNA binding sites than there are ParB dimers. Thus, the ParB in the cell must “make a choice” about where in the chromosome it will bind, and presumably ends up in foci around parS because of the high affinity interaction between ParB and its specific site mediated by the HTH domain. This situation cannot possibly be achieved in vitro. In order to observe ParB:DNA binding we must raise [ParB] to ~1 µM concentration, at which point it will always be in a large stoichiometric excess compared to the binding sites in our radiolabelled or immobilised DNA substrates. It is critical that in vitro experiments are interpreted in the light of this issue, which also highlights the importance of recent superresolution imaging approaches for ParB in vivo in order to study ParB networks. We have added a section to the Discussion section to acknowledge this point.

In our MT experiments there is an interplay between force and concentration. The minimum concentration tested in condensation experiments involving ParB^AF^ was 250 nM.

With respect to the suggested experiments using long DNAs including *parS* sites, we do have some preliminary experiments indicating an accumulation of fluorescence around the region of *parS* sites However, we are still working on this and should regard our current data as too preliminary to publish.

7) It is not very clear if the experimental conditions for condensation inhibition are the same here than in the previous authors' published work. Indeed, previous experiments seem to have been performed in buffer without Mg^2+^ and without EDTA if I understand correctly ("100 mM NaCl, 50 mM Tris-HCl or HEPES-KOH pH 7.5, 100 mg/ml BSA and 0.1% Tween 20 (v/v)" according to Materials and methods section from Fisher et al., 2017). Please clarify.

In the previous work (Fisher et al., 2017) all Magnetic Tweezers experiments were performed in a buffer containing 4 mM MgCl2, 100 mM NaCl, 50 mM Tris-HCl or HEPES-KOH pH 7.5, 100 mg/ml BSA and 0.1% Tween 20 (v/v).

All experiments in the present study were performed either including 4 mM MgCl2 or 1 mM EDTA. We have explicitly included this information in the manuscript in the captions of Figure 2, Figure 3, and Figure 5.

**References**

Aris, R. (1956). *On the dispersion of a solute in a fluid flowing through a tube*. Proc. R. Soc. London Ser. A *235*, 67-77.

Fisher, G.L., Pastrana, C.L., Higman, V.A., Koh, A., Taylor, J.A., Butterer, A., Craggs, T., Sobott, F., Murray, H., Crump, M.P., Moreno-Herrero, F., and Dillingham, M.S. (2017). *The structural basis for dynamic DNA binding and bridging interactions which condense the bacterial centromere*. eLife *6*.

Niman, C.S., Beech, J.P., Tegenfeldt, J.O., Curmi, P.M., Woolfson, D.N., Forde, N.R., and Linke, H. (2013). *Controlled microfluidic switching in arbitrary time-sequences with low drag*. Lab on a chip *13*, 2389-2396.

Schwesinger, F., Ros, R., Strunz, T., Anselmetti, D., Guntherodt, H.J., Honegger, A., Jermutus, L., Tiefenauer, L., and Pluckthun, A. (2000). *Unbinding forces of single antibody-antigen complexes correlate with their thermal dissociation rates*. Proc Natl Acad Sci U S A *97*, 9972-9977.

Taylor, G. (1953). *Dispersion of soluble matter in solvent flowing slowly through a tube*. Proc. R. Soc. London Ser. A *219*, 186-203.

**Kinetic Model:**

Let AA dimers represent the free CTD in solution

Let BB dimers represent the hypothesised CTD dimers that bridge FLParBs in the condensate, and against which force is applied in the magnetic tweezers.

Let AB dimers represent the hypothesised CTD-FLParB heterodimers proposed to cap the ParBDNA complexes.

We are interested in the rate of formation of AB dimers upon mixing of AA and BB.

Kinetic scheme for a simple dimer exchange:

AAk1k2A+A (Eq. R1)

BBk3k4B+B (Eq. R2)

ABk5k6A+B(Eq. R3)

Rate equations

dAAdt=−k1×AA+0.5×K2×A×A(Eq. R4)

dAdt=2∗K1∗AA−k2∗A∗A−k6∗A∗B+k5∗AB(Eq. R5)

dBBdt=−k3∗BB∗+0.5∗k4∗B∗B(Eq. R6)

dAdt=2∗K3∗BB−k4∗B∗B−k6∗A∗B+k5∗AB(Eq. R7)

dABdt=−k5∗AB∗+k6∗A∗B(Eq. R8)

We will use units of nM and seconds throughout.

We will assume that kon (i.e. k2, k4 and k6) is diffusion limited in all cases (kon = 1 nM-1 s -1).

We will assume that the free CTD is a very tight dimer (koff = 0.0001 s-1 equivalent to time constant of about 3 hours, and an equilibrium dissociation constant of 0.1 pM)

We will assume in all cases that at time zero there is no free monomer whatsoever. Thus AB can only be produced by dissociation (of AA and BB) followed by re-association (to form AB).

Scenario 1 (an example Berkeley Madonna syntax in shown below)

[A] = 0 nM

[B] = 0 nM

[AA] = 5000 nM

[BB] = 5000 nM

[AB] = 0 nM

k2 = k4 = k6 = 1 nM^-1^ s^-1^

k1 = k2 = k3 = 0.0001 s^-1^

Upon mixing, A and B will exchange to make ~5000 nM AB. The halftime is ~7000 seconds.

**Author response image 4. respfig4:** 

Scenario 2

[A] = 0 nM

[B] = 0 nM

[AA] = 5000 nM

[BB] = 1 nM

[AB] = 0 nM

k2 = k4 = k6 = 1 nM^-1^ s ^-1^

k1 = k2 = k3 = 0.0001 s^-1^

Upon mixing, A and B will exchange to make ~2 nM AB. The halftime is ~7000 seconds.

**Author response image 5. respfig5:** 

Scenario 3

[A] = 0 nM

[B] = 0 nM

[AA] = 5000 nM

[BB] = 5000 nM

[AB] = 0 nM

k2 = k4 = k6 = 1 nM^-1^ s ^-1^

k1 = k2 = 0.0001 s^-1^

k3 = 0.1 s^-1^ (k3 increased by 1000-fold to mimic applied force, note that BB is still 90% dimer at equilibrium with B)

Upon mixing, A and B will exchange to make ~10000 nM AB. The halftime is ~7000 seconds.

**Author response image 6. respfig6:** 

Scenario 4

[A] = 0 nM

[B] = 0 nM

[AA] = 5000 nM

[BB] = 1 nM

[AB] = 0 nM

k2 = k4 = k6 = 1 nM^-1^ s ^-1^

k1 = k2 = 0.0001 s^-1^

k3 = 0.1 s^-1^ (k3 increased by 1000-fold to mimic applied force, note that BB is still 90% dimer at equilibrium with B)

Upon mixing, A and B will exchange to make ~2 nM AB. The halftime is ~10 seconds.

**Author response image 7. respfig7:** 

Syntax for Berkeley Madonna Simulation (Scenario 1 shown)

METHOD RK4

STARTTIME = 0

STOPTIME = 40000

DT = 1

init AA = 5000

init A = 0

init BB = 5000

init B = 0

init AB = 0

d/dt(AA) = -k1*AA + 0.5*k2*A*A

d/dt(A) = 2*k1*AA - k2*A*A - k6*A*B + k5*AB

d/dt(BB) = -k3*BB + 0.5*k4*B*B

d/dt(B) = 2*k3*BB - k4*B*B - k6*A*B + k5*AB

d/dt(AB) = -k5*AB + k6*A*B

k1 = 0.0001

k2 = 1

k3 = 0.0001

k4 = 1

k5 = 0.0001

k6 = 1